# Cytosolic S100A8/A9 promotes Ca²⁺ supply at LFA-1 adhesion clusters during neutrophil recruitment

**Matteo Napoli[1], Roland Immler[1], Ina Rohwedder[1], Valerio Lupperger[2], Johannes Pfabe[3], Mariano Gonzalez Pisfil[1], Anna Yevtushenko[1], Thomas Vogl[4], Johannes Roth[4], Melanie Salvermoser[1], Steffen Dietzel[1], Marjan Slak Rupnik[3], Carsten Marr[2], Barbara Walzog[1], Markus Sperandio[1], Monika Pruenster[1]\***

[1]Walter Brendel Center of Experimental Medicine, Biomedical Center, Institute of Cardiovascular Physiology and Pathophysiology, Ludwig-Maximilians-University, Planegg-Martinsried, München, Germany; [2]Institute of AI for Health, Helmholtz Zentrum München - German Research Center for Environmental Health, Neuherberg, Germany; [3]Center for Physiology and Pharmacology, Medical University of Vienna, Vienna, Austria; [4]Institute of Immunology, University of Muenster, Muenster, Germany

**\*For correspondence:** monika.pruenster@med.uni-muenchen.de

## eLife Assessment

This **important** study investigates the contribution of cytosolic S100A/8 to neutrophil migration to inflamed tissues. The authors provide **convincing** evidence for how the loss of cytosolic S100A/8 specifically affects the ability of neutrophils to crawl and subsequently adhere under shear stress. This study will be of interest in fields where inflammation is implicated, such as autoimmunity or sepsis.

**Abstract** S100A8/A9 is an endogenous alarmin secreted by myeloid cells during many acute and chronic inflammatory disorders. Despite increasing evidence of the proinflammatory effects of extracellular S100A8/A9, little is known about its intracellular function. Here, we show that cytosolic S100A8/A9 is indispensable for neutrophil post-arrest modifications during outside-in signaling under flow conditions in vitro and neutrophil recruitment in vivo, independent of its extracellular functions. Mechanistically, genetic deletion of S100A9 in mice caused dysregulated Ca²⁺ signatures in activated neutrophils resulting in reduced Ca²⁺ availability at the formed LFA-1/F-actin clusters with defective β₂ integrin outside-in signaling during post-arrest modifications. Consequently, we observed impaired cytoskeletal rearrangement, cell polarization, and spreading, as well as cell protrusion formation in *S100a9*⁻ᐟ⁻ compared to wildtype (WT) neutrophils, making *S100a9*⁻ᐟ⁻ cells more susceptible to detach under flow, thereby preventing efficient neutrophil recruitment and extravasation into inflamed tissue.

## Introduction

Neutrophils are the most abundant circulating leukocyte subpopulation in humans and are rapidly mobilized from the bone marrow to the circulation upon sterile inflammation and/or bacterial/viral infection (*Németh et al., 2020*). The interplay between activated endothelial cells and circulating neutrophils leads to a tightly regulated series of events described as leukocyte recruitment cascade (*Ley et al., 2018*). Tissue-derived proinflammatory signals provoke expression of selectins on the

inflamed endothelium that capture free floating neutrophils from the bloodstream by triggering tethering and rolling through interaction with selectin ligands on the neutrophil surface (*Schmidt et al., 2013*). Selectin-mediated rolling allows neutrophils to engage with immobilized chemokines and other proinflammatory mediators such as leucotriene B4 (LTB4), *N*-formylmethionyl-leucyl-phenylalanine (fMLF), and various agonists for Toll-like receptors (TLRs) like TLR2, TLR4, and TLR5, presented on the endothelial surface and resulting in the activation of $\beta_2$ integrins on neutrophils (*Patcha et al., 2004*; *Chung et al., 2014*; *Pruenster et al., 2015*; *Uhl et al., 2016*). High-affinity $\beta_2$ integrin interaction with their corresponding receptors on the endothelium induces downstream outside-in signaling leading to post-arrest modifications such as cell spreading, adhesion strengthening, and neutrophil crawling, critical requirements for successful recruitment of neutrophils into inflamed tissue (*Immler et al., 2018*; *Begandt et al., 2017*). Accordingly, impairment in those steps favors neutrophil detachment under shear flow and re-entry of neutrophils into the bloodstream (*Chung et al., 2014*).

Local regulation of intracellular calcium ($Ca^{2+}$) levels is critical to synchronize rolling, arrest, and polarization (*Immler et al., 2018*; *Dixit and Simon, 2012*). During rolling, neutrophils show only minor $Ca^{2+}$ activity, but a rapid increase in intracellular $Ca^{2+}$ signaling is registered during transition from slow rolling to firm adhesion and subsequent polarization of neutrophils (*Schaff et al., 2008*).

Neutrophil transition from rolling into firm arrest involves conformational changes of the integrin lymphocyte function-associated antigen (LFA-1) into a high-affinity state allowing bond formation with intercellular adhesion molecule-1 (ICAM-1) expressed on inflamed endothelium. Tension on focal clusters of LFA-1/ICAM-1 bonds mediated by shear stress promotes the assembly of cytoskeletal adaptor proteins to integrin tails and mediates $Ca^{2+}$ release-activated channel (CRAC) ORAI-1 recruitment to focal adhesion clusters ensuring high $Ca^{2+}$ concentrations at the 'inflammatory synapse' (*Dixit and Simon, 2012*). Finally, shear stress-mediated local bursts of $Ca^{2+}$ promote assembly of the F-actin cytoskeleton allowing pseudopod formation and transendothelial migration (*Immler et al., 2018*; *Dixit and Simon, 2012*; *Dixit et al., 2012*; *Dixit et al., 2011*; *Schaff et al., 2010*).

S100A8/A9, also known as MRP8/14 or calprotectin, is a member of the EF-hand family of proteins and the most abundant cytosolic protein complex in neutrophils (*Edgeworth et al., 1991*). Secretion of S100A8/A9 can occur via passive release of the cytosolic protein due to cellular necrosis or neutrophil extracellular trap formation (*Pruenster et al., 2016*). Active release of S100A8/A9 without cell death can be induced by the interaction of L-selectin/PSGL-1 with E-selectin during neutrophil rolling on inflamed endothelium (*Pruenster et al., 2015*; *Morikis et al., 2017*; *Jorch et al., 2023*). We have recently shown that E-selectin-induced S100A8/A9 release occurs through an NLRP3 inflammasome-dependent pathway involving GSDMD pore formation. Pore formation is a time-limited and transient process, which is reversed by the activation of the ESCRT-III machinery membrane repair mechanism (*Pruenster et al., 2023*). Once released, the protein acts as an alarmin, exerting its proinflammatory effects on different cell types like endothelial cells, lymphocytes, and neutrophils (*Pruenster et al., 2016*; *Wang et al., 2018*).

In the present study, we focused on the cytosolic function of S100A8/A9 in neutrophils. We demonstrate its unique role in supplying $Ca^{2+}$ at LFA-1 adhesion clusters during neutrophil recruitment thereby orchestrating $Ca^{2+}$-dependent post-arrest modifications, which are critical steps for subsequent transmigration and extravasation of these cells into inflamed tissues.

## Results

### Cytosolic S100A8/A9 promotes leukocyte recruitment in vivo regardless of extracellular S100A8/A9 functions

As demonstrated previously by our group, rolling of neutrophils on inflamed endothelium leads to E-selectin-mediated, NLRP3 inflammasome-dependent, secretion of S100A8/A9 via transient GSDMD pores (*Pruenster et al., 2023*). Released S100A8/A9 heterodimer in turn binds to TLR4 on neutrophils in an autocrine manner, leading to $\beta_2$ integrin activation, slow leukocyte rolling, and firm neutrophil adhesion (*Pruenster et al., 2015*). Interestingly, E-selectin-triggered S100A9/A9 release does not substantially affect the cytosolic S100A8/A9 content. Analysis of S100A8/A9 levels in the supernatants of E-selectin versus Triton X-100-treated neutrophils demonstrated that only about 1–2% of the cytosolic S100A8/A9 content was secreted to the extracellular compartment (*Figure 1A*). In addition, immunofluorescence analysis of the inflamed cremaster muscle tissue confirmed no major

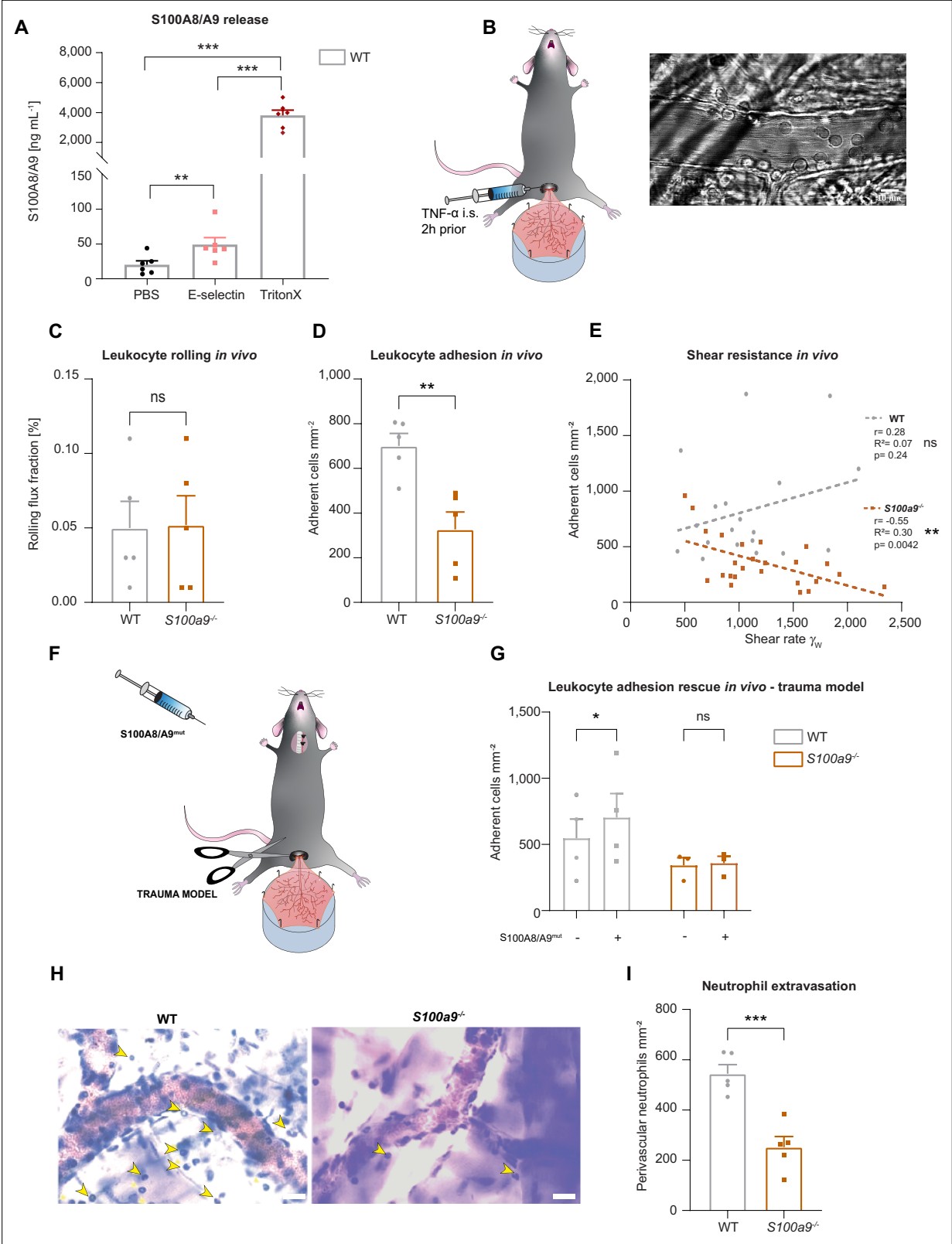

**Figure 1.** Cytosolic S100A8/A9 regulates leukocyte recruitment in vivo regardless of extracellular S100A8/A9. (**A**) Enzyme-linked immunosorbent assay (ELISA) measurements of S100A8/A9 levels in supernatants of wildtype (WT) bone marrow neutrophils stimulated for 10 min with PBS, E-selectin, or lysed with Triton X-100 (mean + SEM, n=6 mice per group, RM one-way analysis of variance [ANOVA], Holm-Sidak's multiple comparison). (**B**) Schematic model of the mouse cremaster muscle preparation for intravital microscopy and representative picture of a vessel showing rolling and adherent

*Figure 1 continued on next page*

*Figure 1 continued*

cells. WT and *S100a9*⁻/⁻ mice were stimulated intrascrotally (i.s.) with TNF-α 2 hr prior to cremaster muscle post-capillary venules imaging by intravital microscopy. Quantification of (**C**) number or rolling (rolling flux fraction) and (**D**) number of adherent neutrophils per vessel surface of WT and *S100a9*⁻/⁻ mice (mean + SEM, n=5 mice per group, 25 [WT] and 30 [*S100a9*⁻/⁻] vessels, unpaired Student's t-test). (**E**) Correlation between physiological vessel shear rates and number of adherent neutrophils in WT and *S100a9*⁻/⁻ mice (n=25 [WT] and 30 [*S100a9*⁻/⁻] vessels of 5 mice per group, Pearson correlation). (**F**) Schematic model of sterile inflammation induced by exteriorizing WT and *S100a9*⁻/⁻ cremaster muscles. (**G**) Analysis of number of adherent leukocytes by intravital microscopy before and after S100A8/A9ᵐᵘᵗ intra-arterial injection (mean + SEM, n=3 mice per group, 3 [WT] and 3 [*S100a9*⁻/⁻] vessels, two-way ANOVA, Sidak's multiple comparison). (**H**) Representative Giemsa staining micrographs of TNF-α stimulated WT and *S100a9*⁻/⁻ cremaster muscles (representative micrographs, scale bar = 30 µm, arrows: transmigrated neutrophils) and (**I**) quantification of number of perivascular neutrophils (mean + SEM, n=5 mice per group, 56 [WT] and 55 [*S100a9*⁻/⁻] vessels, unpaired Student's t-test). ns, not significant; *p≤0.05, **p≤0.01, ***p≤0.001.

The online version of this article includes the following figure supplement(s) for figure 1:

**Figure supplement 1.** Cytosolic S100A8/A9 is indispensable for neutrophil recruitment in vivo.

difference in the amount of cytosolic S100A8/A9 between intravascular and extravasated neutrophils (*Figure 1—figure supplement 1A and B*). Given the abundance of cytosolic S100A8/A9 even after its active release during neutrophil rolling, we wanted to investigate a putative role of intracellular S100A8/A9 in leukocyte recruitment independently of its extracellular function.

To investigate this, we made use of wildtype (WT) and *S100a9*⁻/⁻ mice, which are functional double knockout mice for S100A8 and S100A9 (MRP8 and MRP14) at the protein level, as validated by western blot by our group and others (*Pruenster et al., 2023*; *Manitz et al., 2003*), and studied neutrophil recruitment in mouse cremaster muscle venules upon TNF-α treatment (*Figure 1B*), a well-established model to assess neutrophil recruitment into inflamed tissue in vivo (*Immler et al., 2022*). Two hours after onset of inflammation, we exteriorized the cremaster muscle and investigated the number of rolling and adherent cells by intravital microscopy. While rolling was not affected by the absence of S100A8/A9 (*Figure 1C*), we detected a reduced number of adherent neutrophils in post-capillary cremaster muscle venules of *S100a9*⁻/⁻ compared to WT mice (*Figure 1D*). We found a significant negative correlation between increasing shear rates and the number of adherent leukocytes in *S100a9*⁻/⁻ animals while this correlation could not be detected in WT mice (*Figure 1E*). These findings indicate that lack of cytosolic S100A8/A9 impairs shear stress resistance of adherent neutrophils in vivo. To exclude differences in surface expression of rolling and adhesion relevant molecules on neutrophils, we performed FACS analysis and could not detect differences in the baseline expression of CD11a, CD11b, CD18, CD62L, PSGL1, CXCR2, and CD44 in WT and *S100a9*⁻/⁻ neutrophils (*Figure 1—figure supplement 1C-J*). In order to test whether the observed phenotype of decreased neutrophil adhesion in *S100a9*⁻/⁻ mice was simply a consequence of the lack of extracellular S100A8/A9-induced β₂ integrin activation, we again performed intravital microscopy in the exteriorized but otherwise unstimulated mouse cremaster muscles. In this scenario, only a mild inflammation is induced which leads to the mobilization of pre-stored P-selectin from Weibel-Pallade bodies, but no upregulation of E-selectin and therefore no E-selectin-induced S100A8/A9 release (*Pruenster et al., 2015*). After exteriorization and trauma-induced induction of inflammation in the cremaster muscle tissue, we systemically injected soluble S100A8/A9 via a carotid artery catheter to induce TLR4-mediated integrin activation and firm leukocyte adhesion in exteriorized cremaster muscle venules (*Figure 1F*; *Pruenster et al., 2015*). To prevent S100A8/A9 tetramerization in plasma, which would abolish binding of S100A8/A9 to TLR4 (*Russo et al., 2022*), we took advantage of a mutant S100A8/A9 protein (S100A8/A9ᵐᵘᵗ, aa exchange N70A and E79A) which is unable to tetramerize upon Ca²⁺ binding thereby inducing substantial TLR4 downstream signaling (*Vogl et al., 2018*; *Leukert et al., 2006*). Injection of S100A8/A9ᵐᵘᵗ induced a significant increase in leukocyte adhesion in WT mice (*Figure 1G*), whereas induction of adhesion was completely absent in *S100a9*⁻/⁻ mice (*Figure 1G*), suggesting that loss of S100A8/A9 causes an intrinsic adhesion defect, which cannot be rescued by application of extracellular S100A8/A9 and subsequent TLR4-mediated β₂ integrin activation. In addition, similar results were obtained in the TNF-α stimulated cremaster muscles model (*Figure 1—figure supplement 1K*) where S100A8/A9ᵐᵘᵗ increased leukocyte adhesion in WT mice, but again could not induce an increase in leukocyte adhesion in *S100a9*⁻/⁻ mice (*Figure 1—figure supplement 1L*). In addition, microvascular parameters were quantified in order to compare different vessels in every in vivo experiment and no difference was detected (*Table 1*).

**Table 1.** Microvascular parameters in vivo.

Number of mice, number of vessels, vessel diameter, centerline velocity, wall shear rate and white blood cell (WBC) of TNF-α stimulated wildtype (WT) and S100a9⁻/⁻ mice, as well as of WT and S100a9⁻/⁻ mice treated with mutS100A8/A9 without any prior stimulation (trauma model) and also of TNF-α stimulated WT and S100a9⁻/⁻ mice treated with mutS100A8/A9 (mean + SEM; unpaired Student's t-test).

| | Mice (n) | Venules (n) | Diameter (µm) | Centerline velocity (µm s⁻¹) | Wall shear rate (s⁻¹) | WBC (µl⁻¹) |
|---|---|---|---|---|---|---|
| WT +TNF-α | 5 | 21 | 32.50+0.50 | 1130+50 | 920+50 | 3580+450 |
| *S100a9⁻/⁻*+TNF-α | 5 | 20 | 34+1.5 | 1320+50 | 1010+50 | 3520+200 |
| | | | ns. (p=0.6065) | ns. (p=0.1534) | ns. (p=0.6091) | ns. (p=0.9723) |
| WT +mutS100 A8/A9 | 4 | 4 | 30+5 | 2030+300 | 1800+420 | 5720+800 |
| *S100a9⁻/⁻*+mutS100 A8/A9 | 3 | 3 | 30+2.5 | 1540+350 | 1300+350 | 5460+650 |
| | | | ns. (p=0.8359) | ns. (p=0.3416) | ns. (p=0.3898) | ns. (p=0.7979) |
| WT +TNF-α+mutS100 A8/A9 | 4 | 18 | 31+3 | 1330+330 | 1450+400 | 4100+1000 |
| *S100a9⁻/⁻*+TNF-α+mutS100 A8/A9 | 5 | 24 | 30+3 | 1700+130 | 1500+300 | 3900+500 |
| | | | ns. (p=0.6470) | ns. (p=0.8341) | ns. (p=0.3947) | ns. (p=0.6677) |

Further, we wanted to investigate whether reduced adhesion results in impaired leukocyte extravasation in *S100a9⁻/⁻* mice and stained TNF-α stimulated cremaster muscles of WT and *S100a9⁻/⁻* mice with Giemsa and analyzed number of perivascular neutrophils. Indeed, we observed a reduced number of transmigrated neutrophils in *S100a9⁻/⁻* compared to WT mice (**Figure 1H and I**). However, when we performed transwell experiments upon CXLC1 stimulation under static conditions we found no difference in transmigration between WT and *S100a9⁻/⁻* neutrophils, indicating that S100A8/A9 facilitates neutrophil transmigration in the presence of shear stress but is dispensable for transmigration under static conditions (**Figure 1—figure supplement 1M**). Taken together, these data indicate that cytosolic S100A8/A9 regulates key processes during neutrophil recruitment into inflamed tissue in vivo.

## Loss of cytosolic S100A8/A9 impairs neutrophil adhesion under flow conditions without affecting β₂ integrin activation

Next, we focused on the adhesion defect of S100A8/A9 deficient neutrophils. For this purpose, we used an autoperfused microflow chamber system as described earlier (**Frommhold et al., 2008**). Flow chambers were coated with E-selectin, ICAM-1, and CXCL1 (**Figure 2A**). This combination of recombinant proteins mimics the inflamed endothelium and allows studying leukocyte adhesion under flow conditions. In line with our in vivo findings, lack of S100A8/A9 did not affect leukocyte rolling (**Figure 2B**), but resulted in a lower number of adherent *S100a9⁻/⁻* leukocytes compared to WT leukocytes (**Figure 2C**), without affecting white blood cell (WBC) count (**Table 2**). In line with our in vivo results, additionally coating of flow chambers with extracellular S100A8/A9 induced a slight increase in adhesion of WT neutrophils but not of *S100a9⁻/⁻* neutrophils (**Figure 2D**). Reduced neutrophil adhesion could be a consequence of defective β₂ integrin activation induced by chemokines or other inflammatory mediators (**Chung et al., 2014**; **Pruenster et al., 2015**; **Uhl et al., 2016**; **Abram and Lowell, 2009**). In order to study the effect of S100A8/A9 deficiency on rapid β₂ integrin activation via Gαi-coupled signaling (inside-out signaling), we investigated the capacity of WT and *S100a9⁻/⁻* neutrophils to bind soluble ICAM-1 upon CXCL1 stimulation using flow cytometry (**Figure 2E**). CXCL1 induced a significant and similar increase in soluble ICAM-1 binding in both, WT and *S100a9⁻/⁻* neutrophils (**Figure 2F**), suggesting that Gαi-coupled integrin activation is independent of cytosolic S100A8/A9. To corroborate this finding, we performed a static adhesion assay where we plated WT and *S100a9⁻/⁻* neutrophils on ICAM-1 coated plates, stimulated them with PBS or CXCL1 and quantified the number of adherent cells. As expected, CXCL1 stimulated WT cells displayed increased adhesion to ICAM-1 coated plates compared to PBS treatment (**Figure 2G**). In line with the findings from the soluble

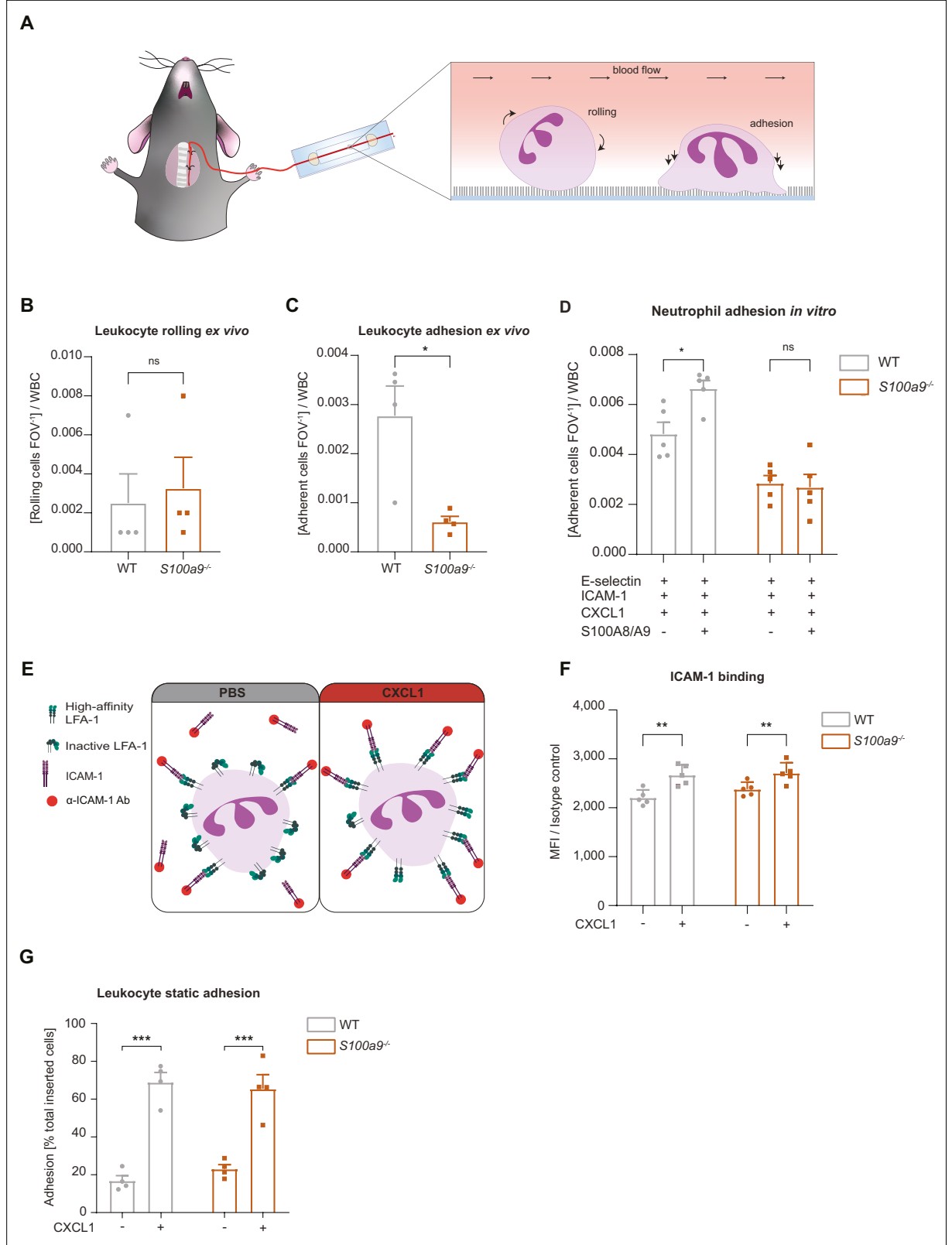

**Figure 2.** Loss of cytosolic S100A8/A9 impairs neutrophil adhesion under flow conditions without affecting $\beta_2$ integrin activation. (**A**) Schematic representation of blood harvesting from wildtype (WT) and *S100a9*[-/-] mice via a carotid artery catheter and perfusion into self-made flow cambers coated with E-selectin, ICAM-1, and CXCL1. Analysis of (**B**) number of rolling and (**C**) number of adherent leukocytes FOV[-1] (mean + SEM, n=4 mice per group, 10 [WT] and 12 [*S100a9*[-/-]] flow chambers, paired Student's t-test). (**D**) Number of adherent leukocytes FOV[-1] in self-made flow chambers coated with

*Figure 2 continued on next page*

*Figure 2 continued*

E-selectin, ICAM-1, CXCL1, and additionally with extracellular S100A8/A9 (mean + SEM, n=5 mice per group, ≥12 [WT] and 14 [*S100a9⁻ᐟ*] flow chambers, two-way analysis of variance (ANOVA), Sidak's multiple comparison). (**E**) Schematic representation of the soluble ICAM-1 binding assay using bone marrow neutrophils stimulated with PBS control or CXCL1 (10 nM) assessed by (**F**) flow cytometry (MFI = median fluorescence intensity, mean + SEM, n=5 mice per group, two-way ANOVA, Sidak's multiple comparison). (**G**) Spectroscopy fluorescence intensity analysis of percentage of adherent WT and *S100a9⁻ᐟ* neutrophils, seeded for 5 min on ICAM-1 coated plates and stimulated with PBS or CXCL1 (10 nM) for 10 min (mean + SEM, n=4 mice per group, two-way ANOVA, Sidak's multiple comparison). ns, not significant; *p≤0.05, **p≤0.01, ***p≤0.001.

ICAM-1 binding assay, this increase was also detected in *S100a9⁻ᐟ* cells indicating that chemokine-induced $\beta_2$ integrin activation is not dependent on cytosolic S100A8/A9.

## Cytosolic S100A8/A9 is crucial for neutrophil spreading, crawling, and post-arrest modifications under flow

Activated and ligand bound $\beta_2$ integrins start to assemble focal clusters thereby transmitting signals into the inner cell compartment (*Legate et al., 2009*). This process named outside-in signaling is required to strengthen adhesion and to induce cell shape changes, fundamental for neutrophil spreading, crawling, and finally transmigration (*Zarbock and Ley, 2009*). Since *S100a9⁻ᐟ* neutrophils displayed a defect in leukocyte adhesion in vivo and ex vivo, although their inside-out signaling is fully functional, we started to study a putative role of cytosolic S100A8/A9 in $\beta_2$ integrin-dependent outside-in signaling. Therefore, isolated WT and *S100a9⁻ᐟ* bone marrow neutrophils were introduced into E-selectin, ICAM-1, and CXCL1 coated microflow chambers and changes in cell shape were monitored over 10 min (*Figure 3A*). WT neutrophils displayed normal spreading properties as depicted by the gradual increase in area and perimeter over time (*Figure 3B*). In line with these findings, circularity and solidity, parameters reflecting the polarization capability of the cells and the amount of protrusions the cell developed, respectively, decreased over time (*Figure 3C*). In contrast, increment of area and perimeter was significantly less pronounced in *S100a9⁻ᐟ* cells (*Figure 3B*). Circularity and solidity did only marginally decrease over time in *S100a9⁻ᐟ* cells, suggesting that neutrophils are unable to polarize properly and to extend protrusions (*Figure 3C*). These results imply a substantial role of cytosolic S100A8/A9 in $\beta_2$ integrin outside-in signaling.

Next, we wanted to examine consequences of impaired neutrophil spreading in absence of S100A8/A9 by analyzing neutrophil crawling under flow. Therefore, we introduced isolated neutrophils into E-selectin, ICAM-1, and CXCL1 coated microflow chambers and allowed them to adhere for 3 min to the substrates. Thereafter, we applied physiological shear stress (2 dyne cm⁻²) and analyzed crawling behavior. WT neutrophils resisted shear forces and slowly crawled in the direction of the flow, whereas *S100a9⁻ᐟ* neutrophils crawled in an intermittent and jerky manner (*Figure 3D* and *Figure 3— video 1*). In line, *S100a9⁻ᐟ* neutrophils covered significantly longer distances (*Figure 3E*), with an increased directionality toward flow direction (*Figure 3F*) and displayed an increased crawling velocity compared to WT cells (*Figure 3G*).

To confirm impaired crawling and defective outside-in signaling-dependent adhesion strengthening in neutrophils lacking cytosolic S100A8/A9, we conducted a neutrophil detachment assay using E-selectin, ICAM-1, and CXCL1 coated microflow chambers and applied increasing shear stress. We found lower numbers of adherent *S100a9⁻ᐟ* neutrophils compared to WT neutrophils with increasing shear stress (*Figure 3—figure supplement 1*). This is in line with our in vivo findings where we detected a negative correlation between the number of adherent cells and increasing shear stress in

**Table 2.** Microvascular parameters ex vivo.
Number of mice, number of flow chambers, cells per field of view (FOV), and white blood cell (WBC) of ex vivo flow chamber assay (mean + SEM, unpaired Student's t-test).

| | Mice (n) | Flow chambers (n) | Cells FOV–1 | WBC(μl–1) |
|---|---|---|---|---|
| WT | 4 | 8 | 39+5 | 8630+1,200 |
| S100a9-/- | 4 | 10 | 37+5 | 8600+1,200 |
| | | | ns. (p=0.7332) | ns. (p=0.9772) |

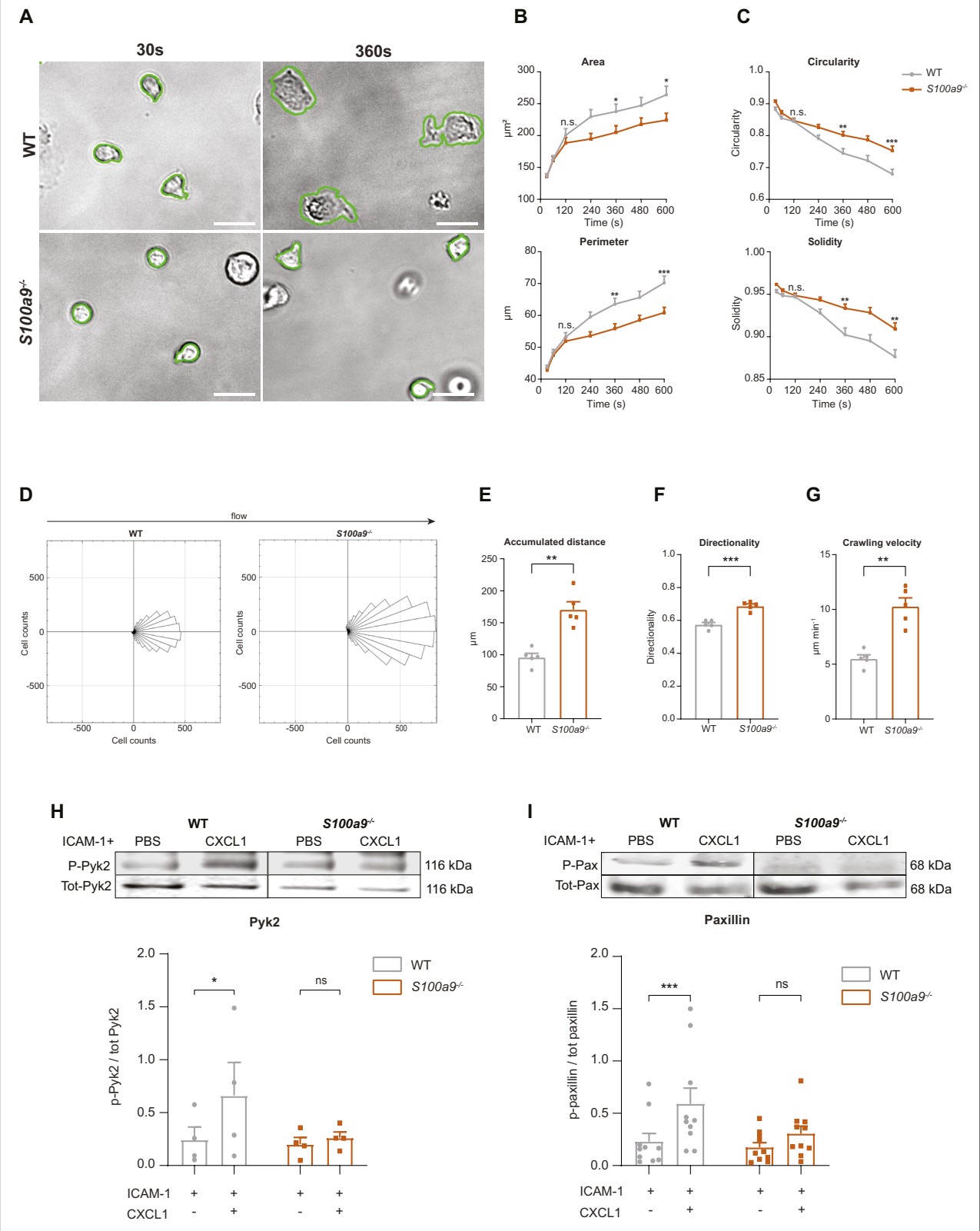

**Figure 3.** Cytosolic S100A8/A9 is crucial for neutrophil spreading, crawling, and post-arrest modifications under flow. (**A**) Representative bright-field pictures of wildtype (WT) and *S100a9*[-/-] neutrophils spreading over E-selectin, ICAM-1, and CXCL1 coated glass capillaries (scale bars = 10 μm). Analysis of cell shape parameters (**B**) area, perimeter, (**C**) circularity (4π * [area/perimeter]) and solidity (area/convex area) over time (mean + SEM, n=103 [WT] and 96 [*S100a9*[-/-]] neutrophils of 4 mice per group, unpaired Student's t-test). (**D**) Rose plot diagrams representative of migratory crawling trajectories of

*Figure 3 continued on next page*

*Figure 3 continued*

WT and *S100a9*[-/-] neutrophils in flow chambers coated with E-selectin, ICAM-1, and CXCL1 under flow (2 dyne cm$^{-2}$). Analysis of (**E**) crawling distance, (**F**) directionality of migration, and (**G**) crawling velocity of WT and *S100a9*[-/-] neutrophils (mean + SEM, n=5 mice per group, 113 [WT] and 109 [*S100a9*[-/-]] cells, paired Student's t-test). Western blot analysis of ICAM-1-induced (**H**) Pyk2 and (**I**) paxillin phosphorylation of WT and *S100a9*[-/-] neutrophils upon CXCL1 stimulation (10 nM) (mean + SEM, representative western blot of n≥4 mice per group, two-way analysis of variance (ANOVA), Sidak's multiple comparison). ns, not significant; *p≤0.05, **p≤0.01, ***p≤0.001.

The online version of this article includes the following video, source data, and figure supplement(s) for figure 3:

**Source data 1.** Pyk2 and paxillin western blot data for *Figure 3H and I*.

**Source data 2.** Pyk2 and paxillin western blot data for *Figure 3H and I*.

**Figure 3—video 1.** Neutrophil functional crawling depends on S100A8/A9.

https://elifesciences.org/articles/96810/figures#fig3video1

**Figure supplement 1.** S100A8/A9 deficient cells are more susceptible to increasing shear stress compared to WT cells.

*S100a9*[-/-] animals (*Figure 1E*). Together, these findings suggest a critical role of intracellular S100A8/A9 in adherent neutrophils to resist high shear stress conditions.

Following engagement of the ligand ICAM-1 to activated β$_2$ integrins in neutrophils, the proline-rich tyrosine kinase Pyk2 and the focal adhesion adaptor protein paxillin are, among other proteins, rapidly tyrosine phosphorylated thereby being critical events for cell adhesion, migration, and podosome formation (*Abram and Lowell, 2009*; *Lu et al., 2022*; *Bouti et al., 2020*). To test the role of cytosolic S100A8/A9 in mediating outside-in signaling events on the mechanistic level, we seeded WT and *S100a9*[-/-] neutrophils on ICAM-1 coated plates, stimulated the cells with CXCL1 and determined Pyk2 and paxillin phosphorylation by western blot analysis. We found increased abundance of Pyk2 and paxillin phosphorylation in CXCL1 stimulated WT cells, while no increase was detectable in *S100a9*[-/-] neutrophils (*Figure 3H and I*). Taken together, these data indicate that cytosolic S100A8/A9 is essential during ICAM-1-induced integrin outside-in signaling events and therefore indispensable for post-arrest modifications including cell polarization and the formation of cell protrusions.

## Cytosolic S100A8/A9 drives neutrophil cytoskeletal rearrangement by regulating LFA-1 nanocluster formation and Ca$^{2+}$ availability within the clusters

Integrin outside-in signaling strongly depends on focal cluster formation of high-affinity LFA-1 and high Ca$^{2+}$ concentrations within these clusters (*Dixit and Simon, 2012*; *Dixit et al., 2012*; *Dixit et al., 2011*; *Schaff et al., 2010*). Since S100A8/A9 is a Ca$^{2+}$ binding protein, we studied LFA-1 clustering and Ca$^{2+}$ signatures during neutrophil adhesion under flow conditions. For this approach, we isolated neutrophils from Ca$^{2+}$ reporter mice (*Lyz2xGCaMP5*) and S100A8/A9 deficient Ca$^{2+}$ reporter mice (*Lyz2xGCaMP5xS100a9*[-/-]) and fluorescently labeled the cells with an LFA-1 antibody (*Figure 4A*). Neutrophils were then introduced into E-selectin, ICAM-1, and CXCL1 coated flow chambers, allowed to settle for 3 min before shear was applied (2 dyne cm$^{-2}$). Time-lapse movies of fluorescence LFA-1 and Ca$^{2+}$ signals were recorded for 10 min by confocal microscopy. First, LFA-1 signals from single-cell analysis (*Figure 4A*) were segmented through automatic thresholding in order to generate a binary image of the LFA-1 signals (LFA-1 mask) (*Figure 4B*). Then, LFA-1 nanoclusters were considered as such if they spanned a minimum area of 0.15 µm$^2$ (*Figure 4C*), according to literature (*Fan et al., 2016*). We found that *Lyz2xGCaMP5xS100a9*[-/-] neutrophils formed significantly less LFA-1 nanoclusters compared to *Lyz2xGCaMP5* neutrophils suggesting an involvement of cytosolic S100A8/A9 in LFA-1 nanocluster formation (*Figure 4D* and *Figure 4—video 1*). Next, we investigated Ca$^{2+}$ intensities within LFA-1 nanoclusters (*Figure 4E*) to determine Ca$^{2+}$ levels at the LFA-1 focal adhesion spots (*Figure 4F*). We found a significant reduction of Ca$^{2+}$ levels in LFA-1 nanocluster areas of *Lyz2xGCaMP5xS100a9*[-/-] neutrophils compared to *Lyz2xGCaMP5* neutrophils (*Figure 4G* and *Figure 4—video 2*), suggesting an impaired availability of free intracellular Ca$^{2+}$ at LFA-1 nanocluster sites in absence of cytosolic S100A8/A9. Strikingly, Ca$^{2+}$ levels in the cytoplasm (outside of LFA-1 nanoclusters, *Figure 4H and I*) did not differ between *Lyz2xGCaMP5* and *Lyz2xGCaMP5xS100a9*[-/-] neutrophils (*Figure 4J*), suggesting that cytosolic S100A8/A9 plays an important role especially in supplying Ca$^{2+}$ at LFA-1 adhesion spots. To investigate localization of S100A8/A9 during neutrophil post-arrest modification, we isolated neutrophils from WT mice and labeled them with the CellTracker Green CMFDA

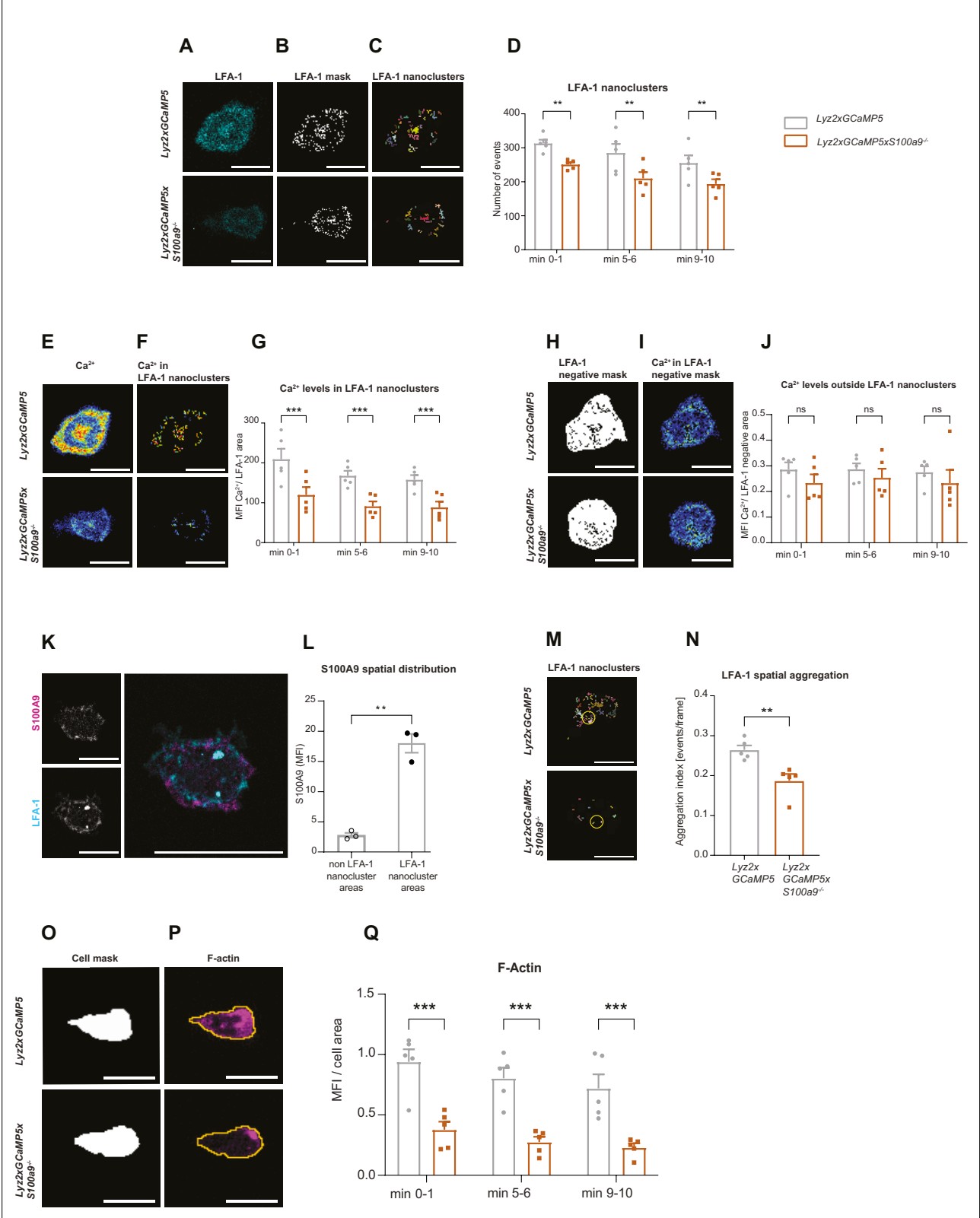

**Figure 4.** Cytosolic S100A8/A9 drives neutrophil cytoskeletal rearrangement by regulating LFA-1 nanocluster formation and Ca²⁺ availability within the clusters. (**A**) Representative confocal images of LFA-1 staining in *Lyz2xGCaMP5* and *Lyz2xGCaMP5xS100a9⁻/⁻* crawling neutrophils on E-selectin, ICAM-1, and CXCL1 coated flow chambers (scale bar = 10 μm). (**B**) Segmentation of LFA-1 signals through automatic thresholding (scale bars = 10 μm). (**C**) Size-excluded LFA-1 nanoclusters of 0.15 μm² minimum size from previously thresholded images (scale bars = 10 μm). (**D**) Single-cell analysis of

*Figure 4 continued on next page*

*Figure 4 continued*

average number of LFA-1 nanoclusters in min 0–1, 5–6, and 9–10 of analysis of *Lyz2xGCaMP5* and *Lyz2xGCaMP5xS100a9*[-/-] neutrophils (mean + SEM, n=5 mice per group, 56 [WT] and 54 [*S100a9*[-/-]] neutrophils, two-way analysis of variance (ANOVA), Sidak's multiple comparison). (**E**) Representative confocal images of Ca$^{2+}$ signals in *Lyz2xGCaMP5* and *Lyz2xGCaMP5xS100a9*[-/-] neutrophils (scale bars = 10 µm) and (**F**) Ca$^{2+}$ signals in the previously segmented LFA-1 nanoclusters (scale bars = 10 µm). (**G**) Quantification of subcellular Ca$^{2+}$ levels in the LFA-1 nanocluster area in min 0–1, 5–6, and 9–10 in *Lyz2xGCaMP5* and *Lyz2xGCaMP5xS100a9*[-/-] neutrophils (mean + SEM, n=5 mice per group, 56 [WT] and 54 [*S100a9*[-/-]] cells, two-way ANOVA, Sidak's multiple comparison). (**H**) Segmented LFA-1 cluster negative areas (scale bars = 10 µm) and (**I**) representative confocal images of Ca$^{2+}$ signals in the LFA-1 cluster negative areas (scale bars = 10 µm) (E–I, scale bar color code: 0=black, 255=white). (**J**) Analysis of cytosolic Ca$^{2+}$ levels in the LFA-1 cluster negative areas in min 0–1, 5–6, and 9–10 of *Lyz2xGCaMP5* and *Lyz2xGCaMP5xS100a9*[-/-] neutrophils (mean + SEM, n=5 mice per group, 56 [WT] and 54 [*S100a9*[-/-]] neutrophils, two-way ANOVA, Sidak's multiple comparison). (**K**) Representative confocal images showing S100A9 localization at LFA-1 nanocluster areas in stimulated WT neutrophils (scale bars = 10 µm). (**L**) Quantitative analysis of S100A9 levels in positive LFA-1 nanocluster areas compared to non-LFA-1 nanocluster areas in stimulated WT neutrophils. (mean + SEM, n=3 mice, 26 [WT] neutrophils, paired Student's t-test). (**M**) Representative confocal micrographs of LFA-1 nanocluster spatial aggregation in *Lyz2xGCaMP5* and *Lyz2xGCaMP5xS100a9*[-/-] neutrophils, within 10 µm$^2$ area and minimum 10 LFA-1 nanoclusters considered (≥10 LFA-1 nanoclusters within 10 µm$^2$, yellow circles = spatial aggregation area, scale bars = 10 µm). (**N**) Analysis of spatially aggregated LFA-1 nanoclusters of *Lyz2xGCaMP5* and *Lyz2xGCaMP5xS100a9*[-/-] neutrophils (mean + SEM, n=5 mice per group, 56 [WT] and 54 [*S100a9*[-/-]] cells, unpaired Student's t-test). (**O**) Segmentation of *Lyz2xGCaMP5* and *Lyz2xGCaMP5xS100a9*[-/-] neutrophil area through *Lyz2* channel automatic thresholding (scale bars = 10 µm) and (**P**) representative confocal images of respective F-actin signals (scale bars = 10 µm). (**Q**) Analysis of F-actin intensity normalized to the cell area in min 0–1, 5–6, and 9–10 of *Lyz2xGCaMP5* and *Lyz2xGCaMP5xS100a9*[-/-] neutrophils (mean + SEM, n=5 mice per group, 74 [WT] and 66 [*S100a9*[-/-]] cells, two-way ANOVA, Sidak's multiple comparison). ns, not significant; *p≤0.05, **p≤0.01, ***p≤0.001.

The online version of this article includes the following video, source data, and figure supplement(s) for figure 4:

**Figure supplement 1—source data 1.** Calmodulin and β-actin western blot data for *Figure 4—figure supplement 1*.

**Figure supplement 1—source data 2.** Calmodulin and β-actin western blot data for *Figure 4—figure supplement 1*.

**Figure 4—video 1.** S100A8/A9 is essential for LFA-1 nanocluster formation and turnover.

https://elifesciences.org/articles/96810/figures#fig4video1

**Figure 4—video 2.** S100A8/A9 increases Ca2+ levels at the LFA-1 nanocluster sites.

https://elifesciences.org/articles/96810/figures#fig4video2

**Figure 4—video 3.** S100A8/A9 induces F-actin polymerization.

https://elifesciences.org/articles/96810/figures#fig4video3

**Figure supplement 1.** S100A8/A9 deficient cells display higher frequencies but shorter duration of Ca$^{2+}$ waves compared to WT cells.

and an LFA-1 antibody. The cells were introduced into flow chambers coated with E-selectin, ICAM-1, and CXCL1, allowed to settle for 3 min, and then subjected to continuous shear stress (2 dyne cm$^{-2}$) for 10 min. After fixation and permeabilization, cells were stained for intracellular S100A9. LFA-1 nanoclusters were identified, and S100A9 intensity in these clusters was compared to that in cytoplasmic areas outside the nanoclusters. We observed higher S100A9 intensity at LFA-1 nanoclusters compared to the rest of the cytoplasm (non LFA-1 nanoclusters) in stimulated WT neutrophils (*Figure 4K and L*), indicating that S100A8/A9 accumulated at LFA-1 nanocluster sites, where it might be critical for local Ca$^{2+}$ supply.

In line, overall Ca$^{2+}$ levels under basal conditions (poly-L-lysine coating, static conditions) were similar between *Lyz2xGCaMP5* and *Lyz2xGCaMP5xS100a9*[-/-] neutrophils (*Figure 4—figure supplement 1A*). Calmodulin levels did not differ between *Lyz2xGCaMP5* and *Lyz2xGCaMP5xS100a9*[-/-] cells as analyzed by western blot (*Figure 4—figure supplement 1B*).

LFA-1 is known to be rapidly recycled and to spatially redistribute to form a ring like structure that co-clusters with endothelial ICAM-1 during neutrophil migration (*Shaw et al., 2004*). To study spatial distribution of LFA-1 nanoclusters (*Figure 4M*), we used Ripley's K function in *Lyz2xGCaMP5* and *Lyz2xGCaMP5xS100a9*[-/-] neutrophils (*Figure 4—figure supplement 1C*). Ripley's K is a spatial statistic that compares a given point distribution with a random distribution (*Dixon, 2001*). *Lyz2xGCaMP5* neutrophils showed significantly more aggregated LFA-1 nanoclusters within 10 µm$^2$ area, suitable for LFA-1 enriched pseudopods, compared to *Lyz2xGCaMP5xS100a9*[-/-] neutrophils (*Figure 4N*), independent from the total LFA-1 nanocluster number. These results show that in the absence of cytosolic S100A8/A9, LFA-1 nanoclusters are more randomly distributed compared to control and indicate that subcellular redistribution of LFA-1 during migration requires cytosolic S100A8/A9.

Recent work has shown that Ca$^{2+}$ signaling promotes F-actin polymerization at the uropod of polarized neutrophils (*Dixit et al., 2011*). Actin waves in turn are known to be important for membrane

protrusion formation, neutrophil polarization, and firm arrest (*Inagaki and Katsuno, 2017*). Therefore, we examined F-actin dynamics in the presence or absence of cytosolic S100A8/A9. For this, we used the same experimental setting as for the LFA-1 cluster analysis but this time we fluorescently labeled *Lyz2xGCaMP5* and *Lyz2xGCaMP5xS100a9*[-/-] neutrophils for F-actin. We generated a mask using the myeloid cell marker *Lyz2* (*Figure 4O*) and applied the mask to the F-actin channel (*Figure 4P*). In line with our previous results on reduced Ca$^{2+}$ levels within LFA-1 adhesion clusters in the absence of S100A8/A9, we found a strongly reduced F-actin signal in *Lyz2xGCaMP5xS100a9*[-/-] neutrophils compared to *Lyz2xGCaMP5* neutrophils (*Figure 4Q* and *Figure 4—video 3*). Total actin levels as determined by western blot analysis did not differ between *Lyz2xGCaMP5* and *Lyz2xG-CaMP5xS100a9*[-/-] neutrophils (*Figure 4—figure supplement 1D*).

Finally, we analyzed the frequency of Ca$^{2+}$ flickers in *Lyz2xGCaMP5* and *Lyz2xGCaMP5xS100a9*[-/-] neutrophils induced by E-selectin, ICAM-1, and CXCL1 stimulation using high-throughput computational analysis. We found an increased number of Ca$^{2+}$ flickers min$^{-1}$ in the absence of S100A8/A9 (*Figure 4—figure supplement 1E and F*), going along with a shorter duration of the Ca$^{2+}$ event compared to control cells (*Figure 4—figure supplement 1G and H*) . This finding suggests that cytosolic S100A8/A9 is not only important for local Ca$^{2+}$ supply at focal LFA-1 nanocluster sites, but in addition 'stabilizes' Ca$^{2+}$ signaling, preventing fast and uncontrolled Ca$^{2+}$ flickering.

Taken together, these data show that cytosolic S100A8/A9 is indispensable for LFA-1 nanocluster formation and actin-dependent cytoskeletal rearrangements by providing and/or promoting Ca$^{2+}$ supply at the LFA-1 nanocluster sites.

## Cytosolic S100A8/A9 is dispensable for chemokine-induced ER store Ca$^{2+}$ release and for the initial phase of SOCE

Our data suggest that intracellular S100A8/A9 is a fundamental regulator of cytosolic Ca$^{2+}$ availability within neutrophils during the recruitment process thereby affecting subcellular LFA-1 and actin dynamics and distribution. Finally, we wanted to study any potential impact of cytosolic S100A8/A9 on Ca$^{2+}$ store release and on store-operated Ca$^{2+}$ entry (SOCE) during neutrophil activation by investigating G-protein-coupled receptors (GPCR)-induced Ca$^{2+}$ signaling using flow cytometry. First, we investigated Ca$^{2+}$ release from the endoplasmic reticulum (ER) and therefore performed the experiments in absence of extracellular Ca$^{2+}$. We could not detect any differences in CXCL1-induced ER store Ca$^{2+}$ release between WT and *S100a9*[-/-] cells, indicating that GPCR-induced downstream signaling leading to ER store depletion is not affected by the absence of cytosolic S100A8/A9 (*Figure 5A and B*). In addition, overall basal Ca$^{2+}$ levels (prior to chemokine stimulation) were similar between WT and *S100a9*[-/-] neutrophils (*Figure 5C*).

Next, we wanted to investigate whether the absence of cytosolic S100A8/A9 might modify chemokine-induced SOCE. Therefore, we stimulated isolated WT and *S100a9*[-/-] neutrophils with CXCL1 in the presence of extracellular Ca$^{2+}$ (*Figure 5D*). Again, basal Ca$^{2+}$ levels were not different between WT and *S100a9*[-/-] cells (*Figure 5E*). Also CRAC functionality was intact as shown by an identical increase in cytosolic Ca$^{2+}$ amount upon CXCL1 stimulation in WT and *S100a9*[-/-] neutrophils (*Figure 5F*). However, we detected different decay kinetics between WT and *S100a9*[-/-] neutrophils as *S100a9*[-/-] neutrophils displayed a steeper decay (*Figure 5G*). Taken together, these data suggest that the presence of cytosolic S100A8/A9 is not a prerequisite for chemokine/GPCR-induced Ca$^{2+}$ release from ER stores and for the initialization of SOCE via CRAC. However, absence of cytosolic S100A8/A9 might disturb Ca$^{2+}$ signaling in a temporal manner.

## Discussion

S100A8/A9 is a Ca$^{2+}$ binding protein, mainly located within the cytosolic compartment of myeloid cells (*Pruenster et al., 2015*; *Pruenster et al., 2016*). Once secreted, S100A8/A9 heterodimers exhibit proinflammatory effects by engagement with its respective receptors including TLR4 and RAGE on a broad spectrum of effector cells, among them phagocytes, lymphocytes, and endothelial cells (*Pruenster et al., 2016*; *Wang et al., 2018*). In addition, extracellular S100A8/A9 is a well-established biomarker for many acute and chronic inflammatory disorders, including cardiovascular diseases, autoimmune diseases, and infections (*Pruenster et al., 2016*; *Wang et al., 2018*; *Jukic et al., 2021*). The tetrameric form of S100A8/A9 was recently shown to have an anti-inflammatory effects during

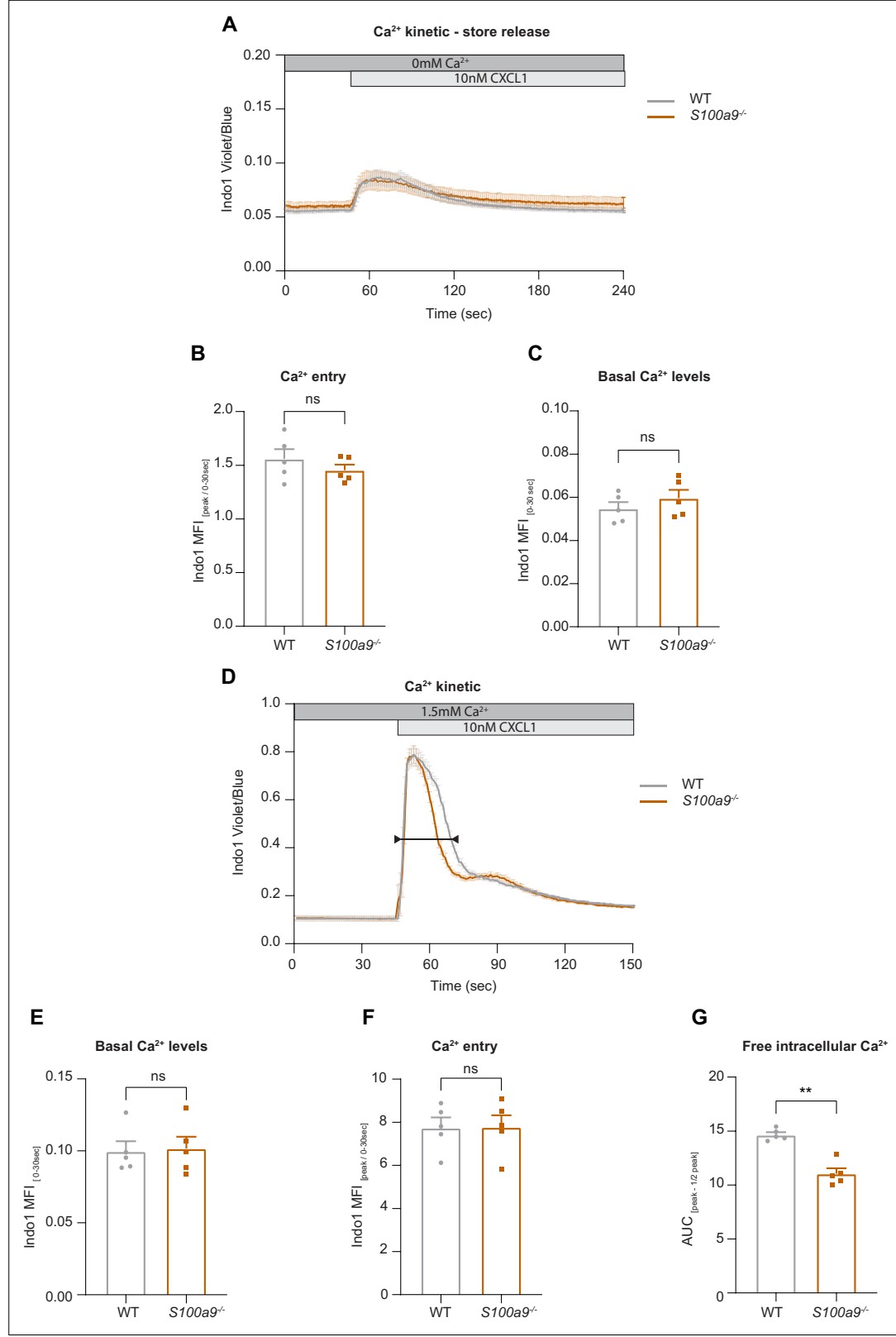

**Figure 5.** Cytosolic S100A8/A9 is dispensable for chemokine-induced endoplasmic reticulum (ER) store Ca²⁺ release and for the initial phase of store-operated Ca²⁺ entry (SOCE). (**A**) Average flow cytometry kinetic graphs of Ca²⁺ store release in the absence of extracellular Ca²⁺ (Ca²⁺ free medium) in wildtype (WT) and *S100a9⁻/⁻* neutrophils upon CXCL1 stimulation (traces are shown as mean + SEM, n=5 mice per group). (**B**) Rapid ER store Ca²⁺ release

*Figure 5 continued on next page*

*Figure 5 continued*

(MFI $_{peak}$/ MFI $_{0-30s}$) of WT and *S100a9$^{-/-}$* neutrophils (mean + SEM, n=5 mice per group, paired Student's t-test). (**C**) Quantification of Ca$^{2+}$ levels under baseline conditions (MFI $_{0-30s}$) (mean + SEM, n=5 mice per group, paired Student's t-test). (**D**) Average flow cytometry kinetic graphs of Ca$^{2+}$ influx in the presence of extracellular Ca$^{2+}$ (HBSS medium, 1.5 mM Ca$^{2+}$) of WT and *S100a9$^{-/-}$* neutrophils upon CXCL1 stimulation (traces are shown as mean + SEM, n=5 mice per group, double-headed arrow represents the time points of quantification). (**E**) Ca$^{2+}$ levels before CXCL1 stimulation (MFI $_{0-30s}$) (mean + SEM, n=5 mice per group, paired Student's t-test). (**F**) Quantification of ER store Ca$^{2+}$ release and calcium release-activated channel (CRAC) store-operated Ca$^{2+}$ entry (MFI $_{peak}$/ MFI $_{0-30s}$) (mean + SEM, n=5 mice per group, paired Student's t-test). (**G**) Ca$^{2+}$ influx after CXCL1 stimulation, from peak to peak half-life (AUC$_{peak - \frac{1}{2}peak}$) of WT and *S100a9$^{-/-}$* neutrophils (mean + SEM, n=5 mice per group, paired Student's t-test). ns, not significant; *p≤0.05, **p≤0.01, ***p≤0.001.

an inflammatory process potentially protecting the organism from overwhelming immune responses (*Russo et al., 2022*). Despite increasing evidence of the pro- and anti-inflammatory effects of secreted S100A8/A9, little is known about its intracellular role in myeloid cells. Here, we show that S100A8/A9 is still abundantly present in the cytosolic compartment of neutrophils even after its active release during inflammation. In addition, we demonstrate that cytosolic S100A8/A9 has a functional impact on neutrophil recruitment during β$_2$ integrin outside-in signaling events by ensuring high Ca$^{2+}$ levels at LFA-1 cluster sites independent of its extracellular functions. Neutrophil β$_2$ integrin outside-in signaling is known to mediate post-arrest modifications including cytoskeletal rearrangements (*Immler et al., 2022*; *Rohwedder et al., 2020*). *S100a9$^{-/-}$* cells, which also lack MRP8 in mature cells of the myeloid lineage (functional S100A8/A9 deficient cells) (*Manitz et al., 2003*; *Ehrchen et al., 2009*), were unable to properly spread, polarize, and crawl. This resulted in a marked impairment of adherent neutrophils to withstand physiological shear forces exerted by the circulating blood. Defective outside-in signaling in absence of S100A8/A9 was accompanied by reduced phosphorylation of paxillin and Pyk2, two critical factors involved in the regulation of β$_2$ integrin mediated cytoskeletal rearrangements (*Abram and Lowell, 2009*). Of note, *S100a9$^{-/-}$* myeloid cells have been shown to comprise alterations of cytoskeletal function before (*Manitz et al., 2003*; *Vogl et al., 2004*; *Roth et al., 1993*; *Goebeler et al., 1995*; *Lominadze et al., 2005*; *Wolf et al., 2023*). In the original publication describing the phenotype of *S100a9$^{-/-}$* mice, an abnormally polarized cell shape of MRP14 deficient cells was described and therefore a potential role of S100A8/A9 in cytoskeletal reorganization was already hypothesized (*Manitz et al., 2003*). In 2014, Vogl et al. demonstrated that cytosolic S100A8/A9 had an impact on the stabilization of microtubules (MTs) via direct interaction of S100A8/A9 with tubulin in resting phagocytes. Upon p38 MAPK and concomitant Ca$^{2+}$ signaling, S100A8/A9 was shown to dissociate from MTs, leading to de-polymerization of MTs thereby allowing neutrophils to transmigrate into inflamed tissue. This might also explain decreased migration of *S100a9$^{-/-}$* granulocytes in a mouse wound healing model (*Vogl et al., 2004*). Additional studies described cytosolic S100A8/A9 to translocate to the membrane and colocalize with vimentin in monocytes upon activation (*Roth et al., 1993*), to interact with keratin in epithelial cells (*Goebeler et al., 1995*) and to associate with F-actin localized to lamellipodia in fMLF stimulated neutrophils (*Lominadze et al., 2005*). Those findings led us to investigate a potential role of cytosolic S100A8/A9 in the Ca$^{2+}$-dependent interplay of plasma membrane located adhesion sites and the cytoskeleton during neutrophil recruitment.

Neutrophil activation during leukocyte recruitment goes along with Ca$^{2+}$ flux initiated e.g. by the engagement of chemokines with GPCRs. Subsequently, phospholipase C beta is activated and leads to the production of inositol-1,4,5-triphosphate (IP3), which in turn elicits the IP3-receptor in the ER, resulting in a rapid Ca$^{2+}$ release from ER stores into the cytoplasm. The decrease in Ca$^{2+}$ concentration in the ER in turn activates the Ca$^{2+}$ sensor stromal interaction molecules (STIM1 and STIM2) triggering the entry of extracellular Ca$^{2+}$ through SOCE mainly via the CRAC ORAI-1 and transient receptor potential channels (*Immler et al., 2018*; *Dixit and Simon, 2012*; *Clemens and Lowell, 2015*). ORAI-1 is recruited to adhesion cluster sites ensuring high Ca$^{2+}$ levels at the 'inflammatory synapse' and rapid rise in intracellular Ca$^{2+}$ concentration, which mediates the assembly of cytoskeletal adaptor proteins to integrin tails and allows the onset of pseudopod formation (*Schaff et al., 2010*; *Simon et al., 2009*). The importance of localized Ca$^{2+}$ availability in subcellular domains has also been shown in T-cells during the engagement with antigen presenting cells within the immunological synapse. In T-cells, mitochondria play a central role as Ca$^{2+}$ buffers and as Ca$^{2+}$ conductors that collect cytosolic Ca$^{2+}$ at the entry site (i.e. through open CRAC located at the immunological synapse) and

distribute it throughout the cytosol (*Kummerow et al., 2009*). Here, we show that in neutrophils cytosolic S100A8/A9 colocalizes with LFA-1 during intravascular adhesion where it might act to increase and stabilize $Ca^{2+}$ availability at the LFA-1 nanocluster sites, mediating spatial clustering of LFA-1 and sustained polymerization of F-actin, both essential steps for efficient neutrophil adhesion strengthening. In addition, presence of cytosolic S100A8/A9 stabilizes duration of $Ca^{2+}$ signals within the cells, as WT cells displayed longer frequencies of $Ca^{2+}$ events with less flickers $min^{-1}$, which might in addition be important for the stability of the inflammatory synapse.

As reported earlier, chemokine-induced $Ca^{2+}$ influx through SOCE at the plasma membrane is indispensable for the activation of high-affinity $\beta_2$ integrins (*Schaff et al., 2008*). This early step during leukocyte recruitment (inside-out signaling) was not affected in absence of cytosolic S100A8/A9. In line, *S100a9⁻/⁻* cells displayed similar CXCL1-induced $Ca^{2+}$ fluxes compared to WT cells. $Ca^{2+}$ release from intracellular stores and initial phases of SOCE were fully functional, as shown by flow cytometry of Indo-1 dye loaded neutrophils. These findings are in accordance with a study by Hobbs et al., which also described normal $Ca^{2+}$ influx in S100A8/A9 deficient neutrophils induced by the chemokine MIP-2 (*Hobbs et al., 2003*). However, we found an impact of cytosolic S100A8/A9 in sustaining high $Ca^{2+}$ concentrations, as $Ca^{2+}$ fluxes decreased faster in the absence of S100A8/A9. Whether this faster decrease is mediated through a direct effect of cytosolic S100A8/A9 on SOCE or through a potential buffer capacity of cytosolic S100A8/A9 needs to be further investigated. Hobbs et al. proposed no impact of S100A9 deletion in the recruitment of neutrophils by using a thioglycolate-induced peritonitis model. However, peritoneal neutrophil emigration was shown to be rather independent of LFA-1 (*Mizgerd et al., 1997*; *Buscher et al., 2016*), whereas extravasation into cremaster muscle tissue strongly relies on the $\beta_2$ integrin LFA-1 and integrin clustering (*Wen et al., 2022*).

Taken together, we identified a critical role of cytosolic S100A8/A9 in neutrophil recruitment under shear stress conditions. We show that its absence leads to reduced $Ca^{2+}$ signaling and impaired sustained $Ca^{2+}$ supply at LFA-1 nanocluster sites. Attenuated $Ca^{2+}$ signatures in turn affect $\beta_2$ integrin-dependent cytoskeletal rearrangements and substantially compromises neutrophil recruitment during the inflammatory response. These findings uncover cytosolic S100A8/A9 as a potentially interesting therapeutic target to reduce neutrophil recruitment during inflammatory disorders with unwanted overwhelming neutrophil influx.

## Materials and methods

### Mice

C57BL/6 WT mice were purchased from Charles Rivers Laboratories (Sulzfeld, Germany). *S100a9⁻/⁻* (functional double S100A8 and S100A9 knockout animals, since the absence of S100A9 also leads to the loss of S100A8 at the protein level; *Manitz et al., 2003*) mice were kindly provided by Johannes Roth (Institute for Immunology, Muenster, Germany). *B6;129S6-Polr2atm1(CAG-GCaMP5g-tdTomato)* crossbred with Lyz2^Cre (*GCaMP5xWT*) were kindly provided by Konstantin Stark (LMU, Munich, Germany) and crossbred with *S100a9⁻/⁻* mice (GCaMP5x *S100a9⁻/⁻*). All mice were housed at the Biomedical Center, LMU Munich, Planegg-Martinsried, Germany. Male and/or female mice (8–25 weeks of age) were used for all experiments. The sample size for animal studies was calculated and optimized based on data from published research or preliminary studies. Experiments were performed in a non-blinded fashion but kept as unbiased as possible. Individual in vivo and in vitro experiments contained appropriate internal controls and normalization methods and were conducted by the same researcher to guarantee reproducibility. Animal experiments were approved by the Regierung von Oberbayern (AZ.: ROB-55.2-2532.Vet_02-17-102 and ROB-55.2-2532.Vet_02-18-22), carried out in accordance with the guidelines from Directive 2010/63/EU and following the ARRIVE guidelines. No mice were excluded. For in vivo experiments, mice were anesthetized via i.p. injection using a combination of ketamine/xylazine (125 and 12.5 mg kg⁻¹ body weight, respectively, in a volume of 0.1 mL NaCl per 8 g body weight). All mice were sacrificed at the end of the experiment by cervical dislocation.

### Neutrophil isolation

Bone marrow neutrophils were isolated using the EasySep Mouse Neutrophil Enrichment Kit according to the manufacturer's instructions (STEMCELL Technologies). Isolated neutrophils were

then resuspended in HBSS buffer (containing 0.1% of glucose, 1 mM $CaCl_2$, 1 mM $MgCl_2$, 0.25% BSA, and 10 mM HEPES [Sigma-Aldrich], pH7.4, complete HBSS).

## S100A8/A9 ELISA

In vitro release of S100A8/A9 was performed as described before (Vogl et al., 2014). Briefly, bone marrow neutrophils were isolated from WT mice. 24-well plates were coated with recombinant murine (rm) E-selectin (rmCD62E-Fc chimera, 10 µg mL$^{-1}$, R&D Systems) or PBS/0.1% BSA at 4°C overnight, blocked with PBS/5% casein (Sigma-Aldrich) and washed twice with PBS. $5\times10^5$ neutrophils were reconstituted in complete HBSS buffer and incubated under shaking conditions on the coated slides for 10 min at 37°C and 5% $CO_2$. To assess the total intracellular S100A8/A9 levels, cells were lysed in 2% Triton X-100 (AppliChem). Finally, cellular supernatants were analyzed by enzyme-linked immuno-sorbent assay (ELISA) to determine the concentrations of S100A8/A9.

## Murine cremaster muscle models

Leukocyte recruitment was investigated by intravital microscopy in inflamed cremaster muscle venules as reported previously (Immler et al., 2020). Shortly, intrascrotal (i.s.) injection of rmTNF-α (500 ng, R&D Systems) was applied to WT and S100a9$^{-/-}$ mice in order to induce an acute inflammation in the cremaster muscle. Two hours after injection, the carotid artery of anesthetized mice was catheterized for later blood sampling (ProCyte Dx; IDEXX Laboratories) or intra-arterial (i.a.) injection. Thereafter, the cremaster muscle was exteriorized and intravital microscopy was conducted on an OlympusBX51 WI microscope, equipped with a ×40 objective (Olympus, 0.8NA, water immersion objective) and a CCD camera (KAPPA CF 8 HS). Post-capillary venules were recorded using VirtualDub software for later analysis. Rolling flux fraction, number of adherent cells mm$^{-2}$, vessel diameter, and vessel length were analyzed using FIJI software (Schindelin et al., 2012). During the entire experiment, the cremaster muscle was superfused with thermo-controlled bicarbonate buffer as described earlier (Ley and Gaehtgens, 1991). Centerline blood flow velocity in each venule was measured with a dual photodiode (Circusoft Instrumentation). Subsequently, cremaster muscles were removed, fixed in 4% PFA solution O.N. at 4°C, and the next day stained with Giemsa (Merck) to assess the number of perivascular neutrophils. The tissues were mounted in Eukytt mounting medium and covered with a 170 µm coverslip. Neutrophils were discriminated from other leukocyte subpopulations based on nuclear shape and granularity of the cytosol. The analysis of transmigrated leukocytes was carried out at the Core Facility Bioimaging of the Biomedical Center with a Leica DM2500 transmission bright-field microscope, equipped with a ×100, 1.4 NA, oil immersion objective, and a Leica DMC2900 CMOS camera. Resulting images had 2048×1536 pixels and a pixel size of 58 nm.

For rescue experiments, we adopted either the TNF-α-induced inflammation model as described above or the trauma-induced inflammation model of the mouse cremaster muscle. In the trauma model, sterile inflammation was induced by opening and exteriorizing the cremaster muscle without application of any stimulus. Intravital microscopy was conducted as described above. After finding an appropriate spot, the same vessel was recorded before and after injection of mutant murine S100A8/S100A9N70AE79A (S100A8/A9$^{mut}$, aa exchange N70A and E79A, 50 µg mouse$^{-1}$ in 100 µL, provided by Thomas Vogl, University of Muenster, Germany) and the number of adherent cells mm$^{-2}$ were counted pre- and post-injection in WT and S100a9$^{-/-}$ mice.

## S100A8/A9 intracellular staining

For the analysis of cytosolic S100A8/A9 levels, TNF-α stimulation of the mouse cremaster muscle was carried out as described above. Subsequently, cremaster muscles were removed, fixed in 4% PFA solution, and immunofluorescence staining for PECAM-1 (Alexa Fluor 488-labeled primary mono-clonal rat antibody, 5 µg mL$^{-1}$, MEC13.3, BioLegend) and S100A9 (Cy5.5 directly labeled, 5 µg mL$^{-1}$, clone 322, provided by Thomas Vogl) was conducted. Stained samples were mounted in Vectashield mounting medium, covered with a 0.17 µm coverslip and imaged by confocal microscopy at the Core Facility Bioimaging of the Biomedical Center, LMU Munich, with an upright Leica SP8X WLL microscope, equipped with an HC PL APO ×40/1.30 NA oil immersion objective. Alexa Fluor 488 was excited with 488 nm, Cy5.5 with 543 nm. Detection windows were 500–568 and 550–640 nm, respectively. Both channels were recorded sequentially. Hybrid photodetectors were used to record images with 512×512 pixels with a pixel size of 0.427 µm. Single-cell analysis was carried out by FIJI software

using macros as follows: MAX projection of Z-stacks were created and neutrophils were segmented by thresholding using S100A8/A9 signal. Then, cell masks were applied back to the original images and S100A8/A9 mean fluorescence intensity (MFI) averaged on stack slices. Finally, S100A8/A9 MFIs were analyzed from intravascular and extravasated neutrophils.

## Neutrophil surface marker staining

Peripheral blood from WT and $S100a9^{-/-}$ mice was harvested and erythrocytes were lysed with lysing solution (BD FACS). Samples were stained for CD18-FITC (5 µg mL$^{-1}$; C71/16; Pharmigen), CD11a-APC (2 µg mL$^{-1}$; M17/4; eBioscience), CD11b-BV510 (0.3 µg mL$^{-1}$; M1/70; BioLegend), CD62L-FITC (5 µg mL$^{-1}$; MEL-14; BioLegend), PSGL1-PE (2 µg mL$^{-1}$; 2PH1; Pharmigen), CXCR2-APC (5 µg mL$^{-1}$; 242216; R&D Systems), CD44-BV570 (0.3 µg mL$^{-1}$; IM7; BioLegend). Respective isotype controls were used: IgG2a-FITC (5 µg mL$^{-1}$; RTK2759; BioLegend), IgG2a-APC (2 µg mL$^{-1}$; RTK2758; BioLegend), IgG2b-BV510 (0.3 µg mL$^{-1}$; RTK4530; BioLegend), IgG1-PE (2 µg mL$^{-1}$; eBRG1; eBioscience), IgG2b-BV570 (0.3 µg mL$^{-1}$; RTK4530; BioLegend). Neutrophils were defined as Ly6G$^+$ cells (0.8 µg mL$^{-1}$; 1A8; BioLegend).

## Neutrophil adhesion ex vivo

Flow chamber assays were carried out as previously described (*Immler et al., 2022*). Briefly, rectangular borosilicate glass capillaries (0.04×0.4 mm$^2$; VitroCom) were coated with a combination of rmE-selectin (CD62E Fc chimera; 20 µg mL$^{-1}$; R&D Systems), rmICAM-1 (ICAM-1 Fc chimera; 15 µg mL$^{-1}$; R&D Systems), and rmCXCL1 (15 µg mL$^{-1}$; Peprotech) for 3 hr at RT and blocked with PBS/5% casein (Sigma-Aldrich) overnight at 4°C. WT and $S100a9^{-/-}$ whole blood was perfused through the microflow chamber either via a carotid artery catheter of anesthetized mice at varying shear stress levels (ex vivo) or via a high-precision pump after being harvested in heparinized tubes (in vitro). Movies were recorded on an OlympusBX51 WI microscope with a ×20, 0.95NA, water immersion objective and a CCD camera (KAPPA CF 8 HS) with VirtualDub software (*Sperandio et al., 2006*). Resulting images had 768×576 pixels and a pixel size of 0.33 µm. Number of rolling and adherent leukocytes/field of view (FOV) were counted using Fiji software, over 1 min time window after 6 min of blood infusion.

## β$_2$ Integrin activation assay

β$_2$ Integrin activation was determined through a modified soluble ICAM-1 binding assay (*Pruenster et al., 2015*). Bone marrow murine neutrophils were isolated as described above. Enriched neutrophils (1.5×10$^6$) were incubated and stained with rmICAM-1 Fc chimera (40 µg mL$^{-1}$, R&D Systems), IgG-Fc-biotin (12.5 µg mL$^{-1}$; eBioscience), and streptavidin-PerCP-Cy5.5 (2 µg mL$^{-1}$; BioLegend). Then, cells were stimulated with rmCXCL1 (10 nM) or PBS (control) in complete HBSS buffer for 5 min at 37°C. The amount of bound rmICAM-1 to the β$_2$ integrin was assessed by flow cytometry (CytoFlex S, Beckmann Coulter) and the median shift relative to the control was analyzed by FlowJo software.

## Static adhesion assay

Neutrophil static adhesion assay was performed as previously described (*Zehrer et al., 2018*). Shortly, 96-well plates were coated with rmICAM-1 (3 µg mL$^{-1}$) overnight at 4°C and washed with PBS. Neutrophils were resuspended in complete HBSS and seeded at 1×10$^5$ cells per well. Cells were allowed to settle for 5 min at 37°C and stimulated with 10 nM rmCXCL1 or PBS (control) for 10 min at 37°C. Using a standard curve, adherent neutrophils were calculated as percentage of total cells added. Standard curve preparation was done by adding 100%, 80%, 60%, 40%, 20%, and 10% of the cell suspension on poly-L-lysine coated wells (100 µg mL$^{-1}$) in triplicates. Non-adherent cells were washed away while adherent cells were fixed with 1% glutaraldehyde and stained with 0.1% crystal violet solution (Sigma-Aldrich). Absorption at 590 nm was measured with a microplate reader (PowerWave HT, Biotek, USA) after lysis of cells with 10% acetic acid solution, as previously described (*Schymeinsky et al., 2009*).

## Spreading assay

To study neutrophil spreading, rectangular borosilicate glass capillaries (0.04×0.40 mm$^2$; VitroCom) were coated with rmE-selectin (CD62E Fc chimera; 20 µg mL$^{-1}$), rmICAM-1 (15 µg mL$^{-1}$), and rmCXCL1 (15 µg mL$^{-1}$) for 3 hr at RT and blocked with PBS/5% casein overnight at 4°C. Bone marrow neutrophils were matured in RPMI 1640 (Sigma-Aldrich) containing FCS (10%, Sigma-Aldrich), GlutaMAX (1%,

Thermo Fisher), Penicillin-Streptomycin solution (1%, Corning) and supplemented with 20% WEHI-3B-conditioned medium overnight at 37°C and applied into the flow chamber at a shear stress level of 1 dyne cm$^{-2}$ using a high-precision syringe pump (Harvard Apparatus, Holliston, MA, USA). Cells were incubated with Fc-block (murine TruStain FcX; BioLegend) for 5 min at RT before being introduced into the chambers. Spreading behavior of the cells was observed and recorded on a Zeiss Axioskop2 with a ×20, 0.5NA water immersion objective and a Hitachi KP-M1AP camera with VirtualDub. Resulting images had 1360×1024 pixels and a pixel size of 600 nm. Cell shape changes were quantified using FIJI software, analyzing cell area, perimeter, circularity ($4\pi \frac{[area]}{[perimeter]^2}$), and solidity ($\frac{[area]}{[convex\ area]}$).

## Crawling assay

15µ-Slides VI$^{0.1}$ (Ibidi) were coated with a combination of rmE-selectin (20 µg mL$^{-1}$), rmICAM-1 (15 µg mL$^{-1}$), and rmCXCL1 (15 µg mL$^{-1}$) for 3 hr at RT and blocked with PBS/5% casein overnight at 4°C. Overnight matured bone marrow neutrophils from WT and *S100a9*$^{-/-}$ mice were resuspended in complete HBSS at 1×10$^6$ mL$^{-1}$, introduced into the chambers and allowed to settle and adhere for 3 min until flow was applied (2 dyne cm$^{-2}$) using a high-precision perfusion pump. Experiments were conducted on a ZEISS, AXIOVERT 200 microscope, provided with a ZEISS ×20 objective (0.25NA), and a SPOT RT ST Camera. MetaMorph software was used to generate time-lapse movies for later analysis. 20 min of neutrophil crawling under flow were analyzed using FIJI software (*Schindelin et al., 2012*) and chemotaxis tool plugin (Ibidi).

## Paxillin and Pyk2 phosphorylation

Paxillin and Pyk2 phosphorylation was investigated as previously described (*Immler et al., 2022*). Briefly, 2×10$^6$ WT or *S100a9*$^{-/-}$ bone marrow murine neutrophils were seeded on rmICAM-1 coated wells (15 µg mL$^{-1}$) for 5 min and stimulated with rmCXCL1 (10 nM) for 5 min at 37°C. Cells were then lysed with lysis buffer (containing 150 mM NaCl, 1% Triton X-100, 0.5% sodium deoxycholate [Sigma-Aldrich], 50 mM Tris-HCl pH7.3 [Merck], 2 mM EDTA [Merck] supplemented with protease [Roche], phosphatase inhibitors [Sigma-Aldrich] and 1x Laemmli sample buffer) and boiled (95°C, 5 min). Cell lysates were resolved by SDS-PAGE and electrophoretically transferred onto PVDF membranes. After subsequent blocking (LI-COR blocking solution), membranes were incubated with the following antibodies for later detection and analysis using the Odyssey CLx Imaging System and Image Studio software: rabbit α-mouse phospho-paxillin (Tyr118) or rabbit α-mouse paxillin and rabbit α-mouse phospho-Pyk2 (Tyr402) or rabbit α-mouse Pyk2 (all Cell Signaling). Goat-α-rabbit IRDye 800RD was used as secondary antibody (LI-COR).

## Detachment assays

To investigate shear resistance, rectangular borosilicate glass capillaries (0.04×0.40 mm$^2$; VitroCom) were coated with rmE-selectin (CD62E Fc chimera; 20 µg mL$^{-1}$), rmICAM-1 (15 µg mL$^{-1}$), and rmCXCL1 (15 µg mL$^{-1}$) for 3 hr at RT and blocked with 5% casein overnight at 4°C. Whole blood from WT and *S100a9*$^{-/-}$ mice was perfused in the coated flow chambers via the cannulated carotid artery, where neutrophils were allowed to attach for 3 min. Then, flow was applied through a high-precision perfusion pump and detachment assays performed over 10 min with increasing shear stress (34–272 dyne cm$^{-2}$) every 30 s. Experiments were recorded by time-lapse movies using the upright Zeiss Axioskop2 with the ×20, 0.5 NA water immersion objective as described above. Number of attached cells was counted at the end of each step.

## Transwell assays

To investigate transmigration under static conditions, WT and *S100a9*$^{-/-}$ neutrophils were seeded on 3 µm tranwell filters and allowed to migrate toward a CXCL1 gradient (10 mM) for 45 min at 37°C. Numbers of transmigrated neutrophils were evaluated using rat anti-Ly6G antibody (1A8; BioLegend), Flow-Count Fluorospheres, and a CytoFlex flow cytometer.

## LFA-1 clustering, S100A8/A9 distribution, Ca$^{2+}$ localization, and F-actin signature during neutrophil crawling under flow

15µ-Slides VI$^{0.1}$ (Ibidi) were used to study LFA-1 clustering, Ca$^{2+}$ localization, and F-actin signature during neutrophil crawling. Flow chambers were coated and blocked as described above. 2×10$^6$

isolated neutrophils from *Lyz2xGCaMP5* or *Lyz2xGCaMP5xS100a9*[-/-] were stained with in-house Alexa Fluor 647-labeled (Antibody Labeling Kit, Invitrogen) monoclonal anti LFA-1 rat antibody (5 µg mL$^{-1}$, 2D7, BD Pharmingen) for 10 min prior to the experiment or SiR-actin (200 nM, Spirochrome) O.N., respectively. Cells were seeded in the chambers and allowed to settle for 2 min before flow was applied (2 dyne cm$^{-2}$) using a high-precision perfusion pump. Samples were imaged by confocal microscopy at the core facility Bioimaging of the Biomedical Center with an inverted Leica SP8X WLL microscope, equipped with an HC PL APO ×40/1.30 NA oil immersion objective. Observation was at 37°C. Hybrid photodetectors were used to record images with 512×512 pixels and a pixel size of 0.284 µm. GCaMP5-GFP was excited with 488 nm, Alexa Fluor 647 or SiR-Actin with 633 nm. Detection windows were 498–540 and 649–710 nm, respectively. For movies, one image was recorded every 0.44 s or every 2 s, over 10 min. Automated single-cell analysis was performed using macros with Fiji software, for minute 0–1, minute 5–6, and minute 9–10 of each recording. For the LFA-1 nanocluster analysis, the LFA-1 channel was automatically segmented and ROIs of a minimum size of 0.15 µm$^2$ were considered as LFA-1 nanoclusters, as reported earlier (*Fan et al., 2016*). This represented a minimum size of 2 pixels in our analysis. The number of clusters was averaged for each analyzed time point (min 0–1, min 5–6, and min 9–10). For the subcellular Ca$^{2+}$ analysis at the LFA-1 cluster sites, the LFA-1 segmented channel was applied to the Ca$^{2+}$ channel and Ca$^{2+}$ events in the selected ROIs were determined, normalized to the LFA-1 areas, and averaged over each minute of analysis. For the Ca$^{2+}$ analysis in the negative LFA-1 area, we again adopted semi-automated single-cell analysis and subtracted the LFA-1 mask from the *Lyz2* mask in order to obtain 'LFA-1 cluster negative masks'. Later, the 'LFA-1 cluster negative masks' were applied to the Ca$^{2+}$ channel and Ca$^{2+}$ intensities were measured, normalized to the 'LFA-1 cluster negative masks' and averaged over each minute of analysis. For the analysis of S100A9 distribution at LFA-1 nanocluster areas, WT neutrophils were stained with CellTracker Green CMFDA (10 µM, Invitrogen) for 45 min and in-house Alexa Fluor 647-labeled monoclonal anti-LFA-1 rat antibody (5 µg/mL, 2D7, BD Pharmingen) for 10 min prior to the experiment. The cells were then seeded in chambers and allowed to settle for 3 min, before applying continuous flow (2 dyne cm$^{-2}$) using a high-precision perfusion pump for 10 min. After the flow, the cells were fixed, permeabilized, and stained overnight at 4°C for intracellular S100A9, followed by counterstaining with DAPI. A semi-automated single-cell analysis was performed to measure S100A9 intensity in the LFA-1 nanocluster areas (obtained as described above) and in the negative LFA-1 nanocluster areas (determined using the same procedure but with CellTracker Green as the cell mask).

For the F-actin analysis, the *Lyz2* channel was automatically segmented to obtain a cell mask and applied to the F-actin channel. F-actin intensities were measured and averaged over each minute of analysis as described above.

## Ca$^{2+}$ store release and Ca$^{2+}$ influx measurement – flow cytometry

Ca$^{2+}$ store release and Ca$^{2+}$ influx was analyzed by flow cytometry through an adapted protocol (*Grimes et al., 2020*). WT and *S100a9*[-/-] bone marrow neutrophils (2.5×10$^6$ mL$^{-1}$) were resuspended in PBS and loaded with 3 µM Indo-1 AM (Invitrogen) for 45 min at 37°C. Cells were washed, resuspended in complete HBSS buffer (2.5×10$^6$ mL$^{-1}$), and stained with an anti-Ly6G-APC antibody (1 µg mL$^{-1}$, 1A8, BioLegend) and with the Fixable Viability Dye eFluor 780 (1:1000; eBioscience). Cells (2×10$^5$) were incubated for 2 min at 37°C and 10 nM CXCL1 was placed on the side of the FACS tube in a 2 µL droplet form. The cells were analyzed at the flow cytometry core facility of the biomedical center with a BD LSRFortessa flow cytometer. Samples were recorded for 45 s to establish a baseline. Afterward, CXCL1 stimulation was initiated by tapping the tube with subsequent fall of the drop into the cell suspension while continuously recording Indo-1 AM signals from neutrophils over time. Data were analyzed using FlowJo software. Calcium levels are expressed as relative ratios of fluorescence emission at 375 nm/525 nm (calcium bound/calcium unbound) and Ca$^{2+}$ signatures quantified as AUC of kinetic averages. To measure Ca$^{2+}$ store release only, Ca$^{2+}$ free medium was used.

## Spatial distribution analysis of LFA-1 nanoclusters

To evaluate the spatial distribution of LFA-1 nanoclusters in neutrophils, Ripley's K statistics (*Dixon, 2001*) was calculated for every time point in every experiment with radii between 0.5 and 5.5 µm. For every radius *r*, we calculated the $K(r)$ value as follows:

$$K(r) = N^{-1} \sum_i \sum_{j \neq i} I\ (dist\ (i,j) \leq r)$$

where $i$ and $j$ are two different LFA-1 nanocluster locations, $I$ is the indicator function which is 1 if the content within the parentheses is 'True' and 0 if the content is 'False', and $N$ is a normalization constant. For 'dist', the Euclidean distance was chosen and calculated via the 'pairwise_distances' from sklearn (**Pedregosa, 2011**). The sampling part of Ripley's K statistic was done by drawing random locations as LFA-1 nanocluster events from the cell surface. To make Ripley's K results comparable between different experiments, we normalized $K(r)$ values such that the random sampling upper bound, calculated for every experiment, was set to 1, and the random sampling lower bound was set to –1. Thus, every normalized value between –1 and 1 is within random borders, i.e., not distinguishable from a random spatial distribution. Values above 1 indicate aggregated LFA-1 nanoclusters and values below –1 indicate dispersed LFA-1 nanoclusters. At least 10 LFA-1 nanoclusters were considered for the spatial aggregation analysis. The number of identified aggregated LFA-1 nanoclusters (values above 1) was counted and averaged for every condition, resulting in an aggregation index.

### Frequency and duration of $Ca^{2+}$ oscillations

After the recording, $Ca^{2+}$ mean intensities of the cells were calculated over time, counting each cell as an individual ROI. Data was imported into a previously described custom analysis pipeline for $Ca^{2+}$ imaging data (**Postić et al., 2023a**). Briefly, the $Ca^{2+}$ mean intensities were sequentially filtered according to the standard values of the pipeline, considering only events with a z-score of at least 3 (p<0.01). From those events, a graph was constructed to detect superimposed events. Properties of the events, AUC or half-width, were used in the calculations afterward. The code of the analysis pipeline can be accessed in the corresponding repository on GitHub (**Postić et al., 2023b**).

### Calmodulin and β-actin western blotting

WT and $S100a9^{-/-}$ bone marrow murine neutrophils ($1 \times 10^6$) were isolated as described above and lysed with lysis buffer and boiled (95°C, 5 min). Cell lysates were resolved by SDS-PAGE and electrophoretically transferred onto PVDF membranes. After subsequent blocking (LI-COR blocking solution), membranes were incubated with the following antibodies for later detection and analysis using the Odyssey CLx Imaging System and Image Studio software. Rabbit α-mouse Calmodulin (5 µg mL$^{-1}$, Cell Signaling), rabbit α-mouse β-actin (1 µg mL$^{-1}$, Cell Signaling), and mouse α-mouse GAPDH (1 µg mL$^{-1}$, Merck/Millipore), goat-α-mouse IRDye 680RD, and goat-α-rabbit IRDye800CW-coupled secondary antibodies (1 µg mL$^{-1}$, LI-COR).

### Statistics

Data were generated from at least three independent experiments and are presented as mean + SEM, as cumulative distribution or representative images, as depicted in the figure legends. Group sizes were selected based on experimental setup. Data were analyzed and illustrated using GraphPad Prism 9 software. Statistical tests were performed according to the number of groups being compared. For pairwise comparison of experimental groups, a paired/unpaired Student's t-test and for more than two groups, a one-way or two-way analysis of variance (ANOVA) with either Tukey's (one-way ANOVA) or Sidak's (two-way ANOVA) post hoc test with repeated measurements were performed, respectively. p-Values<0.05 were considered statistically significant and indicated as follows: *p<0.05; *p<0.01; ***p<0.001.

## Acknowledgements

We thank Dorothee Gössel, Anke Lübeck, Sabine D'Avis, Susanne Bierschenk, and Jennifer Troung for excellent technical assistance, and the core facility Flow Cytometry at the Biomedical Center, LMU, Planegg-Martinsried, Germany. This work was supported by the German Research Foundation (DFG) collaborative research grants: SFB914, projects A02 (BW), B01 (MS), B11 (MP), TRR-332 projects C2 (MS), C3 (BW), C7 (TV), B5 (JR), and the TRR-359 project B2 (MS and RI).

## Additional information

### Funding

| Funder | Grant reference number | Author |
|---|---|---|
| Deutsche Forschungsgemeinschaft | SFB914 | Barbara Walzog<br>Markus Sperandio<br>Monika Pruenster |
| Deutsche Forschungsgemeinschaft | TRR-332 | Thomas Vogl<br>Johannes Roth<br>Barbara Walzog<br>Markus Sperandio |
| Deutsche Forschungsgemeinschaft | TRR-359 | Markus Sperandio<br>Roland Immler |

The funders had no role in study design, data collection and interpretation, or the decision to submit the work for publication.

### Author contributions

Matteo Napoli, Conceptualization, Data curation, Formal analysis, Supervision, Validation, Investigation, Visualization, Methodology, Writing – original draft, Project administration, Writing – review and editing; Roland Immler, Ina Rohwedder, Conceptualization, Data curation, Investigation, Methodology; Valerio Lupperger, Johannes Pfabe, Mariano Gonzalez Pisfil, Data curation, Formal analysis; Anna Yevtushenko, Data curation, Formal analysis, Investigation, Methodology; Thomas Vogl, Resources, Investigation, Methodology; Johannes Roth, Conceptualization, Resources; Melanie Salvermoser, Data curation, Investigation, Methodology; Steffen Dietzel, Marjan Slak Rupnik, Carsten Marr, Barbara Walzog, Conceptualization; Markus Sperandio, Conceptualization, Data curation, Supervision, Funding acquisition, Visualization, Methodology, Project administration, Writing – review and editing; Monika Pruenster, Conceptualization, Data curation, Formal analysis, Supervision, Funding acquisition, Validation, Investigation, Visualization, Methodology, Project administration, Writing – review and editing

### Author ORCIDs

Matteo Napoli ![ORCID] https://orcid.org/0000-0002-8197-9374
Johannes Roth ![ORCID] https://orcid.org/0000-0001-7035-8348
Marjan Slak Rupnik ![ORCID] https://orcid.org/0000-0002-3744-4882
Carsten Marr ![ORCID] https://orcid.org/0000-0003-2154-4552
Markus Sperandio ![ORCID] https://orcid.org/0000-0002-7689-3613
Monika Pruenster ![ORCID] https://orcid.org/0000-0001-8573-6866

### Ethics

Animal experiments were approved by the Regierung von Oberbayern (AZ.: ROB-55.2-2532.Vet_02-17-102 and ROB-55.2-2532.Vet_02-18-22) and carried out in accordance with the guidelines from Directive 2010/63/EU. For surgery, mice were anaesthetized via i.p. injection using a combination of ketamine/xylazine (125mg kg-1 and 12.5mg kg-1 body weight, respectively in a volume of 0.1mL NaCl per 8g body weight). All mice were sacrificed at the end of the experiment by cervical dislocation. Every effort and care was made to minimize suffering of the animals.

Reviewer #1 (Public review): https://doi.org/10.7554/eLife.96810.3.sa1
Reviewer #2 (Public review): https://doi.org/10.7554/eLife.96810.3.sa2
Author response https://doi.org/10.7554/eLife.96810.3.sa3

## Additional files

### Supplementary files
• MDAR checklist

## Data availability

All data are deposited with Dryad. Custom codes developed for data analysis and visualization are available on GitHub, (copy archived at *Napoli, 2023*) and *Postić et al., 2023b*.

The following dataset was generated:

| Author(s) | Year | Dataset title | Dataset URL | Database and Identifier |
|---|---|---|---|---|
| Pruenster et al. | 2024 | Cytosolic S100A8/A9 promotes Ca2+ supply at LFA-1 adhesion clusters during neutrophil recruitment | http://dx.doi.org/10.5061/dryad.7d7wm384t | Dryad Digital Repository, 10.5061/dryad.7d7wm384t |

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
