## [Editor Report · eLife Assessment]

This **important** study investigates the contribution of cytosolic S100A/8 to neutrophil migration to inflamed tissues. The authors provide **convincing** evidence for how the loss of cytosolic S100A/8 specifically affects the ability of neutrophils to crawl and subsequently adhere under shear stress. This study will be of interest in fields where inflammation is implicated, such as autoimmunity or sepsis.

---

## [Referee Report · Reviewer #1 (Public review)]

Summary:

In this manuscript by Napoli et al, the authors study the intracellular function of Cytosolic S100A8/A9 a myeloid cell soluble protein that operates extracellularly as an alarmin, whose intracellular function is not well characterized. Here, the authors utilize state-of-the-art intravital microscopy to demonstrate that adhesion defects observed in cells lacking S100A8/A9 (Mrp14-/-) are not rescued by exogenous S100A8/A9, thus highlighting an intrinsic defect. Based on this result subsequent efforts were employed to characterize the nature of those adhesion defects.

Strengths:

The authors convincingly show that Mrp14-/- neutrophils have normal rolling but defective adhesion caused by impaired CD11b activation (deficient ICAM1 binding). Analysis of cellular spreading (defective in Mrp14-/- cells) are also sound. The manuscript then focuses on selective signaling pathways and calcium measurements. Overall, this is a straightforward study of biologically important proteins and mechanisms.

Weaknesses:

Some suggestions are included below to improve this manuscript.

---

## [Referee Report · Reviewer #2 (Public review)]

Summary:

Napoli et al. provide a compelling study showing the importance of cytosolic S100A8/9 in maintaining calcium levels at LFA-1 nano clusters at the cell membrane, thus allowing the successful crawling and adherence of neutrophils under shear stress. The authors show that cytosolic S100A8/9 is responsible for retaining stable and high concentrations of calcium specifically at LFA-1 nanoclusters upon binding to ICAM-1, and imply that this process aids in facilitating actin polymerisation involved in cell shape and adherence. The authors show early on that S100A8/9 deficient neutrophils fail to extravasate successfully into the tissue, thus suggesting that targeting cytosolic S100A8/9 could be useful in settings of autoimmunity/acute inflammation where neutrophil-induced collateral damage is unwanted.

Strengths:

Using multiple complementary methods from imaging to western blotting and flow cytometry, including extracellular supplementation of S100A8/9 in vivo, the authors conclusively prove a defect in intracellular S100A8/9, rather than extracellular S100A8/9 was responsible for the loss in neutrophil adherence, and pinpointed that S100A8/9 aided in calcium stabilisation and retention at the plasma membrane.

Weaknesses:

(1) Extravasation is shown to be a major defect of Mrp14-/- neutrophils, but the Giemsa staining in Figure 1H seems to be quite unspecific to me, as neutrophils were determined by nuclear shape and granularity, which could be affected by the angle at which the nucleus is viewed. It would have perhaps been cleaner/clearer to use immunofluorescence staining for neutrophils instead as seen in Supplementary Figure 1A (staining for Ly6G or other markers instead of S100A9).

Addressed issues:

(1) The representative image for Mrp14-/- neutrophils used in Figure 4K to demonstrate the Ripley's K function seems to be very different from that shown above in Figure 4C and 4F. In their response to reviewers, the authors reassure that all data has been included in the analysis.

(2) In the initial submission the authors needed to provide a more direct linkage between cytosolic S100A8/9 and actin polymerisation, which subsequently results in the arrest and adherence of neutrophils. The authors did an additional experiment indicating the co-localization of S100A8/9 with LFA-1, indicating that the spatial localisation of S100A8/9 does shift towards the membrane with activation. Further, the authors confirm that the defect is only apparent only in conditions of shear stress, as transwell migration of Mrp14-/- neutrophils is not affected.

---

## [Author Response]

The following is the authors’ response to the original reviews.

**Reviewer #1:**
In this manuscript by Napoli et al, the authors study the intracellular function of Cytosolic S100A8/A9 a myeloid cell soluble protein that operates extracellularly as an alarmin, whose intracellular function is not well characterized. Here, the authors utilize state-of-the-art intravital microscopy to demonstrate that adhesion defects observed in cells lacking S100A8/A9 (Mrp14-/-) are not rescued by exogenous S100A8/A9, thus highlighting an intrinsic defect. Based on this result subsequent efforts were employed to characterize the nature of those adhesion defects.

The authors thank reviewer #1 for his/her insightful comments and suggestions. Please find our point to point responses below.

(1) Ex vivo characterization of the function of S100A8/A9 in adhesion, spreading, and calcium signaling requires at least one rescue experiment to support the direct role of these proteins in the biological processes under study.

We thank the reviewer for this comment. We agree that rescue experiments would be helpful to confirm the direct role of intracellular S100A8/A9 in adhesion, spreading, and Ca2+ signaling. Although transfection of primary cells, especially neutrophils, poses challenges due to their short half-life, we now have undertaken additional in vitro rescue experiments. Specifically, we used extracellular S100A8/A9 and coated Ibidi flow chambers with E-selectin, ICAM-1 and CXCL1 alone or alongside S100A8/A9, and measured rolling and adhesion of blood neutrophils. Our data reveal that extracellular S100A8/A9 can induce increased adhesion in WT neutrophils but fails to rescue the adhesion defect in Mrp14-/- neutrophils (Author response image 1). This result corroborates our in vivo findings, emphasizing that the observed adhesion defect is due to the lack of intracellular S100A8/A9.

**Author response image 1. sa3fig1:** Extracellular S100A8/A9 does not rescue the adhesion defect in Mrp14/- neutrophils. Analysis of number of adherent leukocytes FOV-1 normalized to the WBC of WT and Mrp14-/- mice. Whole blood was harvested through a carotid artery catheter and perfused with a high precision pump at constant shear rate using flow cambers coated with either E-selectin, ICAM-1 and CXCL1 or E-selectin, ICMA-1, CXCL1 and S100A8/A9. [mean + SEM, n=5 mice per group, 12 (WT) and 14 (Mrp14-/-) flow chambers, 2way ANOVA, Sidak’s multiple comparison]. ns, not significant; *p≤0.05, **p≤0.01, ***p≤0.001.

(2) There is room for improvement in the analysis of signaling pathways presented in Figures 3 H and I. Western blots and analyses are not convincing, in particular for p-Pax.

We acknowledge the reviewer's concern regarding the clarity of the signaling pathway analysis, particularly the western blots for p-Paxillin. To address this, we have repeated the western blot experiments using murine neutrophils. Our new data confirm the defective paxillin phosphorylation upon CXCL1 stimulation and ICAM-1 binding in the absence of cytosolic S100A8/A9. We have now integrated these new findings with the original data and included the updated results in the manuscript (Figure 3I revised). These enhanced analyses provide a more robust and convincing demonstration of the signaling defects in Mrp14-/- neutrophils.

(3) At least one western blot showing a knockdown of S100A8/A9 should be included towards the beginning of the result section.

We appreciate the reviewer's suggestion to include a western blot demonstrating the knockout of S100A8/A9 early in the results section. In a recent publication by our group, we have already demonstrated the absence of S100A8/A9 at the protein level in Mrp14-/- neutrophils via western blotting ([1], please refer to Extended Data Fig. 1h). We agree that visual confirmation of the absence of S100A8/A9 protein is crucial for establishing the validity of our study.

(4) The Ca2+ measurements at LFA-1 nanoclusters using the Mrp14-/- Lyz2xGCamP5 are interesting; It is understood that the authors are correcting calcium levels by normalizing by LFA-1 cluster areas and that seems fine to me. The issue is that the total calcium signal seems decreased in Mrp14-/- cells compared to WT cells (Fig. 4E)...why is totalCa2+ low? Please discuss.

We thank the reviewer for this insightful comment. Indeed, our observations reveal reduced overall Ca2+ levels in Mrp14-/- neutrophils compared to WT neutrophils. Initially, we noticed a general decrease in Ca2+ intensity (Author response image 2A-B) and lifetime in Mrp14-/- neutrophils (Author response image 2C-D). Further analysis indicated that these differences in Ca2+ levels are localized specifically to the LFA-1 nanocluster sites. In contrast, the cytosolic Ca2+ levels outside of the LFA-1 nanocluster areas were comparable between Mrp14-/- and WT neutrophils (Figure 4H-J). This suggests that the reduced total Ca2+ levels observed in Mrp14-/- neutrophils are primarily due to the impaired Ca2+ supply at the LFA-1 nanocluster areas. Our data support the notion that cytosolic S100A8/A9 plays a crucial role in actively supplying Ca2+ to LFA-1 nanoclusters during neutrophil crawling. In the absence of S100A8/A9, the increase in overall Ca2+ levels (summing both inside and outside LFA-1 nanocluster areas) is minimal, further highlighting the specific role of S100A8/A9 in maintaining localized Ca2+ concentrations at these crucial sites.

**Author response image 2. sa3fig2:** Overall Ca2+ levels in WT and Mrp14-/- neutrophils (A) Representative confocal images of neutrophils from WT Lyz2xGCaMP5 and Mrp14-/- Lyz2xGCaMP5 mice, labeled with Lyz2 td Tomato marker. The images illustrate overall cytosolic Ca2+ levels during neutrophil crawling flow chambers coated with E-selectin, ICAM-1, and CXCL1 (scale bars=10μm). (B) Quantitative analysis of total cytosolic Ca2+ intensity in single cells from WT Lyz2xGCaMP5 and Mrp14-/- Lyz2xGCaMP5 neutrophils measured over three time intervals: min 0-1, 5-6 and 9-10 [mean + SEM, n=5 mice per group, 56 (WT) and 54 (Mrp14-/-) neutrophils, 2way ANOVA, Sidak’s multiple comparison]. (C) Representative traces and (D) single cell analysis of total Ca2+ lifetime over the first 5 minutes in WT Lyz2xGCaMP5 and Mrp14-/- Lyz2xGCaMP5 neutrophils crawling on Eselectin, ICAM-1, and CXCL1 coated flow chambers recorded with FLIM microscopy [mean + SEM, n=3 mice per group, 111 (WT) and 95 (Mrp14-/-) neutrophils, 2way ANOVA, Sidak’s multiple comparison]. ns, not significant; *p≤0.05, **p≤0.01, ***p≤0.001.

(5) Even if the calcium level outside LFA-1 nanoclusters is not significant (Figure 4J), the data at min 9-10 in Figure 4J seems to be affected by a single event that may be an outlier. Additional data may be needed here.

We appreciate the reviewer’s attention to this detail. To address the concern regarding a potential outlier in the Ca2+ level measurements at 9-10 minutes in Figure 4J, we rigorously tested the dataset using the GraphPad outlier calculator. The analysis revealed that no data point was statistically identified as an outlier. Given that the current dataset is robust and the statistical analysis confirms the integrity of the data, we believe that the results accurately reflect the biological variability observed in our experiments. Therefore, we have not added additional data points at this stage but remain open to discussing this further.

(6) Finally, even though there is less calcium at LFA-1 clusters, that does not necessarily mean that "cytosolic S100A8/A9 plays an important role in Ca2+ "supply" at LFA-1 adhesion spots" as proposed. S100A8/A9 may play an indirect role in calcium availability. The analysis of the subcellular localization of S100A8/A9 at LFA-1 clusters together with calcium dynamics in stimulated WT cells would help support the authors' interpretation, which although possibly correct, seems speculative at this point.

We thank the reviewer for this insightful comment and fully agree that additional evidence regarding the subcellular localization of S100A8/A9 would strengthen our conclusions. Although live cell imaging of intracellular S100A8/A9 was initially challenging due to technical limitations, we have now performed additional experiments to address this issue. We conducted end-point measurements where we allowed WT neutrophils to crawl on E-selectin, ICAM-1, and CXCL1 coated flow chambers for 10 minutes. Following this, we fixed and permeabilized the cells to stain intracellular S100A9, along with LFA-1 and a cell tracker for segmentation. Confocal microscopy and subsequent single-cell analysis revealed a significant enrichment of S100A8/A9 at LFA-1 positive nanocluster areas compared to the surrounding cytosol (Figure 4K and 4L, new). This finding supports our hypothesis that S100A8/A9 plays a direct role in the localized supply of Ca2+ at LFA-1 adhesion spots, thus facilitating efficient neutrophil crawling under shear stress. These new data have been included in the revised manuscript, providing stronger evidence for our proposed mechanism.

**Reviewer #2:**
Napoli et al. provide a compelling study showing the importance of cytosolic S100A8/9 in maintaining calcium levels at LFA-1 nanoclusters at the cell membrane, thus allowing the successful crawling and adherence of neutrophils under shear stress. The authors show that cytosolic S100A8/9 is responsible for retaining stable and high concentrations of calcium specifically at LFA-1 nanoclusters upon binding to ICAM-1, and imply that this process aids in facilitating actin polymerisation involved in cell shape and adherence. The authors show early on that S100A8/9 deficient neutrophils fail to extravasate successfully into the tissue, thus suggesting that targeting cytosolic S100A8/9 could be useful in settings of autoimmunity/acute inflammation where neutrophil-induced collateral damage is unwanted.

The authors appreciate reviewer #2's insightful comments and suggestions. Below are our detailed responses:

(1) Extravasation is shown to be a major defect of Mrp14-/- neutrophils, but the Giemsa staining in Figure 1H seems to be quite unspecific to me, as neutrophils were determined by nuclear shape and granularity. It would have perhaps been more clear to use immunofluorescence staining for neutrophils instead as seen in Supplementary Figure 1A (staining for Ly6G or other markers instead of S100A9).

We acknowledge the reviewer's concern. However, Giemsa staining is a well-established method in hematology, histology, cytology, and bacteriology, widely recognized for its ability to distinguish leukocyte subsets based on nuclear shape and cytoplasmic characteristics. This method is extensively documented in the literature [2-5]. Its advantages are the easy morphological discrimination of leukocytes based on nuclear and cytoplasmic shape and conformation (Author response image 3).

**Author response image 3. sa3fig3:** Giemsa staining of extravasated leukocyte subsets. (A) Representative image of Giemsa-stained cremaster muscle tissue post-TNF stimulation. The image clearly differentiates leukocyte subsets (white arrow = neutrophils, yellow arrow = eosinophils, red arrow = monocytes). Scale bar = 50µm.

(2) The representative image for Mrp14-/- neutrophils used in Figure 4K to demonstrate Ripley's K function seems to be very different from that shown above in Figures 4C and 4F.

The reviewer correctly observed that the cell in Figure 4K is different from those in Figures 4C and 4F. This is intentional, as Figure 4K is meant to show a representative image that accurately reflects the overall results of the experiments. We assure the reviewer that all cells analyzed in Figures 4C and 4F were also included in the analysis for Figure 4K.

(3) Although the authors have done well to draw a path linking cytosolic S100A8/9 to actin polymerisation and subsequently the arrest and adherence of neutrophils in vitro, the authors can be more explicit with the analysis - for example, is the F-actin co-localized with the LFA-1 nanoclusters? Does S100A8/9 localise to the membrane with LFA-1 upon stimulation? Lastly, I think it would have been very useful to close the loop on the extravasation observation with some in vitro evidence to show that neutrophils fail to extravasate under shear stress.

We thank the reviewer for this comment and questions.

Concerning the co-localization of F-actin with LFA-1 nanoclusters and S100A8/9 localization: We appreciate the reviewer's interest in the co-localization between F-actin and LFA-1. Unfortunately, due to the limitations of our GCaMP5 mouse model (with neutrophils labeled with td-Tomato and eGFP for LyzM and Ca2+), we could only stain for either LFA-1 or F-actin at a time. However, in our F-actin movies, we observed that F-actin predominantly localizes at the rear of the cell, while LFA-1 is more uniformly distributed at the plasma membrane.

Regarding S100A8/A9 localization, as mentioned in response to Reviewer 1's sixth point, we now conducted endpoint measurements. We stained neutrophils with cell tracker green CMFDA and LFA-1, allowed them to crawl on E-selectin, ICAM-1, and CXCL1-coated flow chambers, and then performed intracellular S100A9 staining after fixation and permeabilization. Our analysis shows higher S100A9 intensity at LFA-1 positive areas compared to LFA-1 negative areas (Figure 4K and 4L, new). This indicates that S100A8/A9 indeed concentrates Ca2+ at LFA-1 nanoclusters, supporting adhesion and post-arrest modification events under flow.

Regarding the extravasation defect under shear stress: To address the reviewer's suggestion, we performed transwell migration assays under static conditions. Our results show no significant difference in transmigration between WT and Mrp14-/- neutrophils without flow, indicating that the extravasation defect in Mrp14-/- neutrophils is shear-dependent. This supports our hypothesis that S100A8/A9-mediated Ca2+ supply at LFA-1 nanoclusters is critical under flow conditions (Author response image 4).

**Author response image 4. sa3fig4:** Static Transmigration assay. (a) Transmigration of WT and Mrp14-/- neutrophils in static transwell assays (3um pore size, 45min migration time) showing spontaneously migration (PBS) or migration towards CXCL1. [mean + SEM, n=3 mice per group, 2way ANOVA, Sidak’s multiple comparison]. ns, not significant; *p≤0.05, **p≤0.01, ***p≤0.001.

Additional References

(1) Pruenster, M., et al., E-selectin-mediated rapid NLRP3 inflammasome activation regulates S100A8/S100A9 release from neutrophils via transient gasdermin D pore formation. Nature Immunology, 2023. 24(12): p. 2021-2031.

(2) Kuwano, Y., et al., Rolling on E- or P-selectin induces the extended but not high-affinity conformation of LFA-1 in neutrophils. Blood, 2010. 116(4): p. 617-24.

(3) Porse, B., Mouse Hematology – A Laboratory Manual. European Journal of Haematology, 2010. 84(6): p. 554-554.

(4) Frommhold, D., et al., Protein C concentrate controls leukocyte recruitment during inflammation and improves survival during endotoxemia after efficient in vivo activation. Am J Pathol, 2011. 179(5): p. 2637-50.

(5) Braach, N., et al., RAGE Controls Activation and Anti-Inflammatory Signalling of Protein C. PLOS ONE, 2014. 9(2): p. e89422.